# Norovirus evolution in immunodeficient mice reveals potentiated pathogenicity via a single nucleotide change in the viral capsid

Forrest C. Walker[1,☉], Ebrahim Hassan[1,☉], Stefan T. Peterson[1], Rachel Rodgers[1,2], Lawrence A. Schriefer[1], Cassandra E. Thompson[1], Yuhao Li[1], Gowri Kalugotla[1], Carla Blum-Johnston[1], Dylan Lawrence[1], Broc T. McCune[3], Vincent R. Graziano[4], Larissa Lushniak[1], Sanghyun Lee[1], Alexa N. Roth[5], Stephanie M. Karst[5], Timothy J. Nice[6], Jonathan J. Miner[3,7,8], Craig B. Wilen[4]*, Megan T. Baldridge[1,8]*

1 Division of Infectious Diseases, Department of Medicine, Edison Family Center for Genome Sciences & Systems Biology, Washington University School of Medicine, St. Louis, Missouri, United States of America, 2 Department of Pediatrics, Washington University School of Medicine, St. Louis, Missouri, United States of America, 3 Department of Pathology & Immunology, Washington University School of Medicine, St. Louis, Missouri, United States of America, 4 Departments of Laboratory Medicine & Immunobiology, Yale School of Medicine, New Haven, Connecticut, United States of America, 5 Department of Molecular Genetics & Microbiology, Emerging Pathogens Institute, University of Florida, Gainesville, Florida, United States of America, 6 Department of Molecular Microbiology and Immunology, Oregon Health & Science University, Portland, Oregon, United States of America, 7 Division of Infectious Diseases, Department of Medicine, Washington University School of Medicine, St. Louis, Missouri, United States of America, 8 Department of Molecular Microbiology, Washington University School of Medicine, St. Louis, Missouri, United States of America

☉ These authors contributed equally to this work.
* craig.wilen@yale.edu (CBW); mbaldridge@wustl.edu (MTB)

**Data Availability Statement:** Sequencing data have been uploaded to the European Nucleotide

## Abstract

Interferons (IFNs) are key controllers of viral replication, with intact IFN responses suppressing virus growth and spread. Using the murine norovirus (MNoV) system, we show that IFNs exert selective pressure to limit the pathogenic evolutionary potential of this enteric virus. In animals lacking type I IFN signaling, the nonlethal MNoV strain CR6 rapidly acquired enhanced virulence via conversion of a single nucleotide. This nucleotide change resulted in amino acid substitution F514I in the viral capsid, which led to >10,000-fold higher replication in systemic organs including the brain. Pathogenicity was mediated by enhanced recruitment and infection of intestinal myeloid cells and increased extraintestinal dissemination of virus. Interestingly, the trade-off for this mutation was reduced fitness in an IFN-competent host, in which CR6 bearing F514I exhibited decreased intestinal replication and shedding. In an immunodeficient context, a spontaneous amino acid change can thus convert a relatively avirulent viral strain into a lethal pathogen.

## Author summary

The evolution of viruses both within a single host and during transmission between hosts often leads to novel characteristics as mutations develop, with increased lethality or

Archive with accession number PRJEB38177
(ERP121566).

**Funding:** This work was supported by the National
Institutes of Health (NIH) grants R01AI141478 (S.
M.K., M.T.B.), R01AI139314 and R01AI127552 (M.
T.B.), and the Pew Biomedical Scholars Program of
the Pew Charitable Trusts (M.T.B.). F.C.W. was
supported by NIH T32GM007067. B.T.M. was
supported by NIH F31CA177194. C.B.W. was
supported by NIH R01AI48467 and K08AI128043
and a Burroughs Wellcome Fund Career Award for
Medical Scientists. The funders had no role in
study design, data collection and analysis, decision
to publish, or preparation of the manuscript.

**Competing interests:** The authors have declared
that no competing interests exist.

transmissibility being particularly concerning potential outcomes. Here, using a novel *in vivo* experimental evolution strategy with intestinal pathogen murine norovirus (MNoV), we identified a single nucleotide mutation that consistently and exclusively arose when mice lacking interferon signaling, a key aspect of the early innate immune response, were infected intracranially with a nonlethal strain of MNoV. This mutation was both sufficient and necessary for viral virulence, conferring virulence by enhanced infection of immune cells to facilitate systemic spread, representing a shift in the cell types infected by the virus. Conversely, this mutation was also associated with more limited infection in the intestine, suggesting a fitness cost. Our work identifies key immunological constraints on the evolution of virulence and provides specific mechanistic insights into viral pathogenesis.

## Introduction

The investigation of viral evolution has long been vital to deciphering viral pathogenesis. This is particularly true for RNA viruses, which evolve much more rapidly than DNA viruses [1]. For example, HIV develops mutations within days to weeks of infection to evade the immune response [2–4], and, within a single individual, can rapidly develop clinically relevant changes in cell tropism [5]. *In vivo* viral evolution studies with influenza have also provided key insights into the genetic basis of virulence and transmission [6,7]. Recent improvements in deep sequencing technology have made it increasingly accessible to probe inter- and intra-host evolution of viral genomes and have been applied extensively to influenza [8,9], HIV [10–12], and poliovirus [13,14]. Host interferons (IFNs) exert critical selective pressure to control viral infections, thereby regulating both viral evolution and transmission [15,16].

Human norovirus (HNoV), the leading global cause of viral gastroenteritis, is a rapidly evolving RNA virus with extensive genetic diversity, as HNoV strains are spread across at least three of the seven genogroups of noroviruses [17]. The dominant circulating strains of the virus that cause pandemics are replaced approximately every three years due to antigenic drift and shift [18]. HNoV remains a major concern in children, causing a large number of deaths worldwide each year. While it generally causes an acute, self-limiting infection in immunocompetent adults, it can be shed in stool for weeks to years in immunocompromised individuals [19,20]. These chronically infected individuals could be the source of novel variants that cause new epidemics [21,22].

In 2003, the discovery, isolation, and characterization of murine norovirus (MNoV) from the brains of $Rag2^{-/-}Stat1^{-/-}$ mice, which lack key facets of both innate and adaptive immunity, provided a tractable *in vivo* system for norovirus studies [23,24]. MNoV, like HNoV, is a non-enveloped, positive sense, single stranded RNA virus in the family *Caliciviridae*, and has since become the leading small animal model of HNoV [25]. Its use has enabled detailed mechanistic studies of clinically relevant aspects of HNoV infection such as transmission, tropism, and persistence [26].

Although there is substantial genetic diversity between HNoV isolates, there is limited diversity among the numerous reported MNoV strains, all of which are within a single genogroup [17]. In spite of the limited (<13%) nucleotide divergence between MNoV strains, there are large strain-specific phenotypic differences [27–29]. Minor changes in the MNoV genome are capable of imparting large phenotypic changes [28]. For example, two well-studied MNoV molecular clones, CR6 and CW3, which are 95% identical at the amino acid level, exhibit dramatic differences in persistence, tropism, and lethality [24,27,29]. CR6 persists for >70 days post-infection (dpi) predominantly in the proximal colon, exploiting intestinal tuft

cells as an immune-privileged reservoir [27,30,31]. In wild-type (WT) mice, orally-transmitted CR6 is confined to the intestine and draining mesenteric lymph nodes (MLN), as type I IFN prevents viral spread to extraintestinal tissues including the liver and spleen [29,32,33]. Even in $Stat1^{-/-}$ mice, which lack the STAT1 transcription factor required for IFN stimulated gene expression, oral CR6 is nonlethal [29]. In contrast, CW3, an infectious clone of the prototypical MNoV strain MNV-1, is cleared by the adaptive immune system by 14 dpi in WT mice [34,35]. CW3 also exhibits a dramatically different cellular tropism *in vivo*, infecting immune cells including macrophages, neutrophils, dendritic cells, and B and T cells [36–38], and is rapidly lethal to both $Stat1^{-/-}$ and $Ifnar1^{-/-}$ mice, which lack only type I IFN signaling [28,39].

Viral adaptation during *in vitro* passaging has previously identified residues involved in *in vivo* MNoV virulence after as few as three passages, highlighting the ability of experimental evolution to reveal key viral determinants of virulence [29,40,41]. These characteristics make MNoV an ideal model system in which to study viral evolution *in vivo*. One of the key factors for many aspects of MNoV virulence, including lethality and tropism, is the viral capsid. The MNoV capsid is comprised of 180 copies of the VP1 protein, which contains a shell (S) domain and protruding (P) domain [42]. The P domain is further divided into P1 and P2 subdomains; the P2 domain is located on the exterior surface of the capsid, and has the highest sequence variation within VP1. The P2 subdomain directly binds to the proteinaceous MNoV receptor, CD300LF [43–45] and many differential aspects of MNoV virulence have been mapped to VP1 and specifically to the P2 domain [29,32,41,46]. However, the determinants of virulence have only rarely been mapped to specific amino acid residues within VP1, supporting further inquiry to characterize the role of VP1 in MNoV virulence.

As MNoV was isolated in 2003 via intracranial passaging in mice lacking IFN signaling, we hypothesized that the original isolation method may account for specific phenotypic features of MNV-1 and the subsequent infectious clone CW3 [23]. We speculated that intracranial inoculation may expose an endemic, nonlethal virus to distinct selective pressures, resulting in virus evolution characterized by increased lethality in immunocompromised mice. Additionally, passaging in $Stat1^{-/-}$ mice may have also facilitated development of important genetic variants. Here, we demonstrate that after only two rounds of passaging *in vivo*, CR6 acquired a key characteristic of CW3: lethality when administered orally to $Stat1^{-/-}$ mice. Deep sequencing and use of the MNoV reverse genetics system identified a single amino acid change in MNoV VP1 that was sufficient for viral pathogenesis *in vivo*, causing CR6 to phenotypically resemble CW3 in tissue tropism and lethality. Further study revealed that this amino acid change was also necessary for viral spread to the brain and lethality. Our approach revealed a critical role for the capsid in determining key aspects of viral pathogenesis, including cell and tissue tropism and lethality, outside of the well-appreciated importance of the receptor-binding domain in these phenotypes.

## Results

### Intracranial inoculation of $Stat1^{-/-}$ mice with persistent MNoV causes systemic infection

We inoculated six $Stat1^{-/-}$ mice with $10^6$ plaque forming units (PFU) of CR6 intracranially (IC) and collected tissue and stool samples from three mice at 3 days post-infection (dpi), then from the remaining mice either at time of death or up to 30 dpi (**Fig 1A**). In parallel, we inoculated wild-type (WT) mice with the same dose IC, and performed per oral (PO) inoculations of $10^6$ PFU of CR6 in both WT and $Stat1^{-/-}$ mice as controls (**S1A Fig**). From our first round of infections, of the three $Stat1^{-/-}$ mice in which infection was allowed to proceed past 3 dpi, one mouse died at 4 dpi, another at 7 dpi, and the third survived until euthanized at 30 dpi

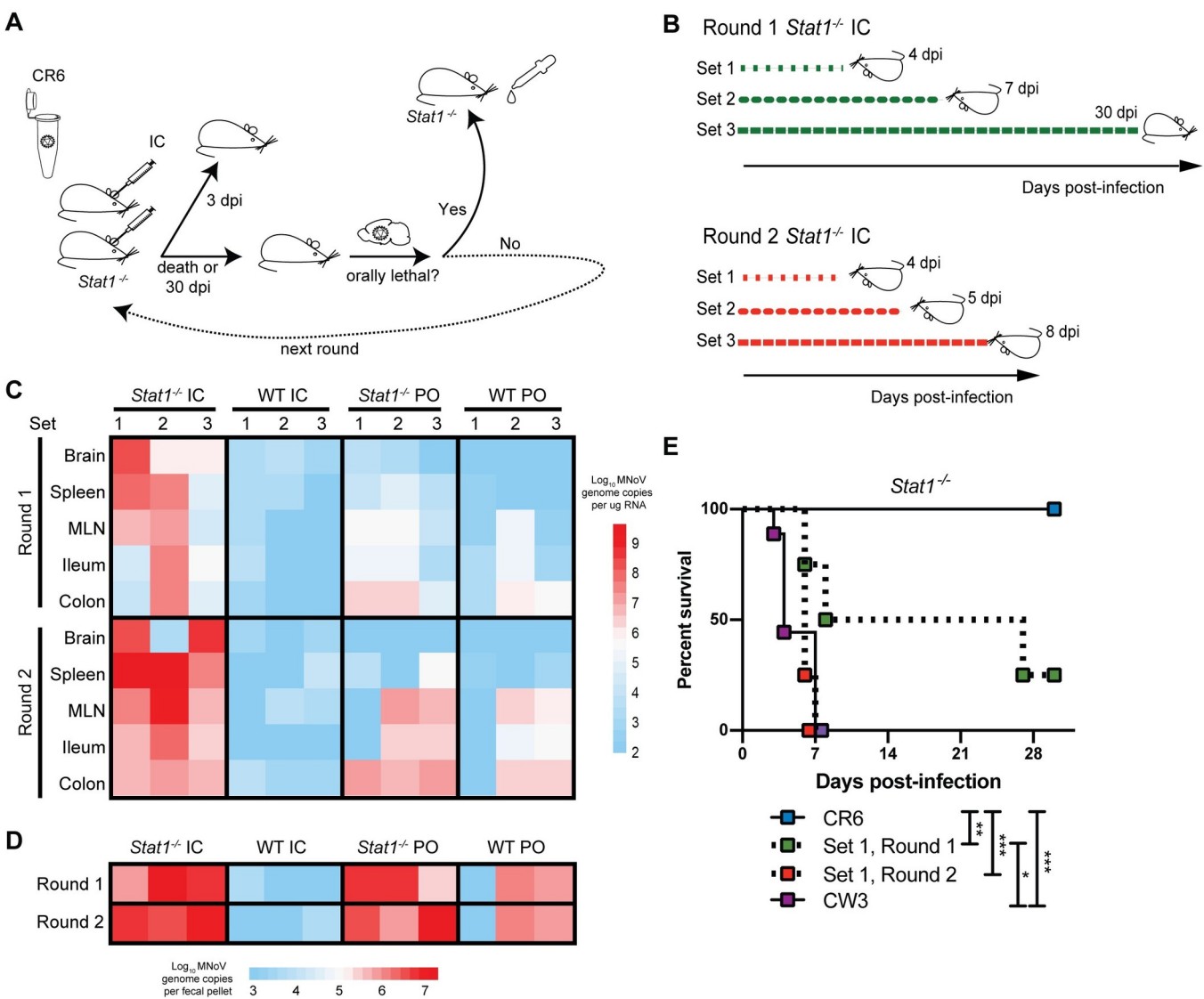

**Fig 1. Intracranial passaging of murine norovirus in *Stat1*[-/-] mice promotes systemic spread and acquisition of lethality. (A,B)** Schematics of intracranial passaging protocol, initiated with $10^6$ PFU CR6, conducted in three parallel "sets" of *Stat1*[-/-] mice **(A)**, and of timing of spontaneous lethality or tissue collection for the three sets of mice **(B)**. **(C,D)** Heatmaps depicting MNoV genome levels detected by quantitative RT-PCR (qPCR) in designated tissues **(C)** or stool **(D)** at time of death as shown in **(B)**, with values ranging from $\log_{10}$ 1.95 to 9.61 (limit of detection at 2.0) for tissues and $\log_{10}$ 2.95 to 7.12 (limit of detection at 3.0) for stool. Each square indicates a single mouse tissue from 24 total mice analyzed. **(E)** Survival curve of *Stat1*[-/-] mice orally administered $10^6$ PFU CW3 or CR6, or a 1:80 dilution of brain homogenate from Set 1 Round 1 or Round 2 mice. *n* = 4–9 mice per group over two independent experiments, analyzed by Mantel-Cox test. ***, *P* < 0.001; **, *P* < 0.01; *, *P* < 0.05; ns, not significant.

(**Fig 1B**). Brains from these mice were harvested and homogenized, and homogenate from each mouse was separately inoculated IC into naive *Stat1*[-/-] mice in parallel for a second round of infection (**Fig 1A and 1B**). Lethality in the second round was accelerated to 4 dpi, 5 dpi, and 8 dpi, respectively (**Fig 1B**). Mice in the control groups (WT IC, WT PO, or *Stat1*[-/-] PO) were euthanized either at 3 dpi or at the time of death of matched experimental mice (**S1A Fig**) to provide time-matched controls. For these control groups, passaging was performed using WT IC brain homogenate for the WT IC group, or fecal material for the WT PO and *Stat1*[-/-] PO groups (**S1A Fig**).

We analyzed viral levels in brain, spleen, mesenteric lymph nodes (MLN), ileum, colon, and stool for all mice harvested at 3 dpi (**S1B Fig**) or at time of death (**Fig 1C and 1D**). Observed trends were broadly similar between Rounds 1 and 2 for each genotype and inoculation route combination. After IC inoculation in $Stat1^{-/-}$ mice, viral genomes disseminated to the spleen by 3 dpi at high levels (average of $\log_{10}$ 6.7 genome copies) and in all tissues examined at later timepoints, indicating broad viral dissemination in the absence of intact IFN signaling. In contrast, limited ($<\log_{10}$ 4.0 genome copies) viral dissemination was observed in WT mice inoculated IC, and WT mice inoculated PO exhibited infection limited to the intestine and MLN, as has been observed consistently with this model [27]. $Stat1^{-/-}$ mice infected PO exhibited robust intestinal infection as well as viral dissemination to the spleen, consistent with the observation that type I IFN signaling limits extraintestinal spread of CR6 [32,33], but dissemination to the brain was limited (**Fig 1C**). Thus, the combination of IC infection and impaired IFN signaling permits CR6-induced viral dissemination and lethality.

## Passaging of virus via intracranial inoculation of $Stat1^{-/-}$ mice confers oral lethality

We next tested whether CR6 virus passaged intracranially in $Stat1^{-/-}$ mice had acquired the capacity to cause lethality in $Stat1^{-/-}$ mice when administered orally, a phenotype observed with CW3 but not CR6 infection [23,47] (**Fig 1E**). We administered brain homogenate from one of three separate IC passages ("Set 1") in $Stat1^{-/-}$ mice, from the first and second rounds, PO to naïve $Stat1^{-/-}$ mice. We found that while the homogenate from the first round attained significantly enhanced lethality compared to CR6, the second round homogenate exhibited 100% lethality, similar to CW3 (**Fig 1E**). Thus, passaging a non-pathogenic strain of MNoV in an IFN signaling-deficient brain conferred increased lethality within one to two rounds of passaging in an immunocompromised setting.

## Single amino acid change in VP1 is detected in parallel passaging

To test whether CR6 accumulated viral mutation(s) that correlated with altered pathogenesis, we developed a method to efficiently deep sequence MNoV genomes (**S2A Fig**). We used a tagmentation protocol to generate sequencing libraries from stool cDNA, then a hybridization-based capture to enrich for viral genomes from these libraries. We compared the percentage of reads mapping to the MNoV genome pre- and post-enrichment using stool cDNA samples from a variety of CR6-infected mice, and found that enrichment improved sequencing efficiency, which was further correlated with viral loads (**S2B and S2C Fig**).

We developed a pipeline to map sequencing reads to visualize coverage depth across and report variants from the CR6 genome. Despite low viral abundance in some samples, this approach yielded nearly complete genomes for many samples and partial genomes for others (mean of 68.5% [range 0–99.2%] genome coverage at a minimum depth of 100 reads/base) (**Figs 2A and S3A**). From this sequencing data, we observed a number of nucleotide positions at which mutations occurred in multiple mice (**S1 Table**). Notably, the majority of these mutations were found within the viral capsid gene VP1, which has previously been identified as a hotspot for mutations in longitudinal sequencing of HNoV strains within chronically-infected immunocompromised patients [48]. When comparing viral variants observed specifically in stool and tissues from $Stat1^{-/-}$ mice inoculated IC to control groups, one mutation in viral capsid gene VP1 at nucleotide position 6595 (T6595A) was observed exclusively in $Stat1^{-/-}$ IC samples (**Fig 2A**). Although we recovered complete or nearly complete genomes from samples with high viral loads, namely $Stat1^{-/-}$ mice as well as WT mice infected orally, some samples lacked adequate sequencing coverage at this position (**S3A Fig**). We therefore used Sanger

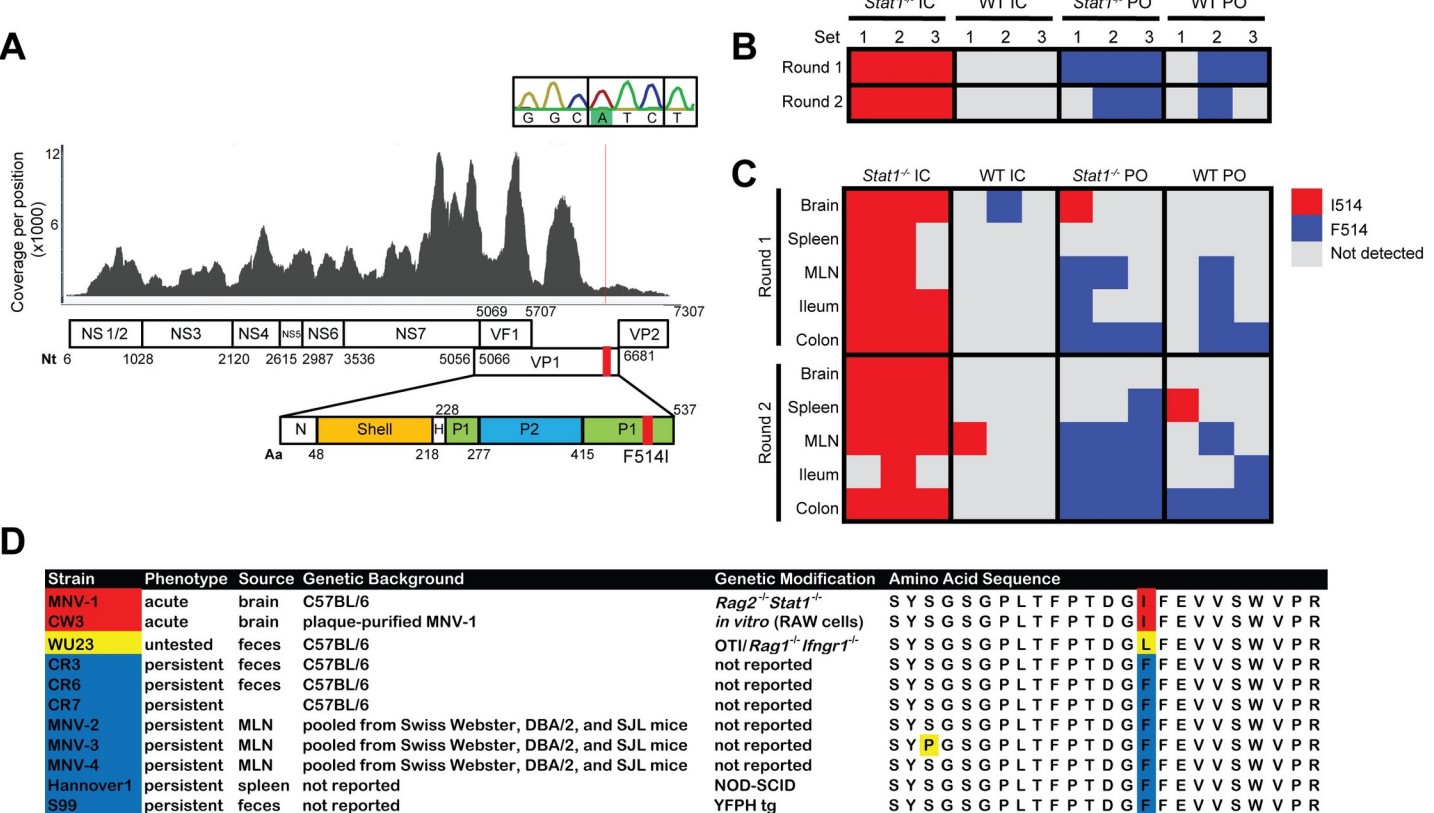

**Fig 2. Amino acid change F514I is observed in *Stat1*⁻/⁻ IC samples and in virulent MNoV strains. (A)** Alignment of "Set 1, Round 2" stool Illumina sequencing reads to the CR6 genome was performed and visualized using an in-house pipeline, with the position 6595 mutation from the reference designated by a red line. Nucleotide (nt) positions along the MNoV genome are depicted along the open reading frames encoding viral proteins. Amino acid (aa) positions for the VP1 protein are depicted below the genome, including N-terminal arm (N) and hinge (H) regions in addition to the shell, P1, and P2 domains. Sanger sequencing-based confirmation of T6595A, encoding F514I, was also performed and is shown in the inset box, with the codon for this amino acid outlined. **(B,C)** Heatmap of Sanger sequencing-based analysis of position 6595 of indicated stools **(B)** and tissues **(C)** at time of death as depicted in **Fig 1B**, with resulting encoded amino acid depicted by color; "not detected" indicates that sequencing was insufficient across this position to call the base. **(D)** Alignment of a subset of VP1 amino acids (positions 500–523) for persistent and acute murine norovirus strains, with position 514 highlighted.

sequencing of this genomic region in all stool samples to confirm that this mutation, which results in a phenylalanine to isoleucine change at amino acid 514 (F514I), was exclusively present in stool samples from *Stat1*⁻/⁻ mice inoculated IC (**Figs 2B and S3B**). Further, it was almost exclusively found in tissue samples from this same group (**Figs 2C and S3C**).

Analysis of the amino acid sequences of VP1 from published MNoV genomes revealed that there was substantial conservation of this protein between persistent and acute MNoV strains, with 88.5% amino acid identity shared between eleven characterized isolates. From these sequences, we additionally observed that there were only ten unique amino acid variants in acute, virulent strain MNV-1 and its infectious clone CW3 that were not observed in any of these nonlethal, persistent strains, including the F514I mutation (**Figs 2D and S3D**). Indeed, among 86 protein sequences for VP1 from original MNoV isolates deposited in the NCBI Protein database, the only published MNoV strains that do not have a phenylalanine at position 514 were MNV-1 and the poorly-characterized strain WU23. Interestingly, a leucine was present at this position in WU23, which, similar to MNV-1, was originally isolated from immuno-compromised mice [28] (**Fig 2D**). Strains SDHD-46 and MNVSH1603, which bear an isoleucine and leucine at position 514, respectively, have not been published aside from

sequence deposition to NCBI, so their *in vivo* phenotypes are unclear. Thus, a single amino acid change in VP1 consistently correlated with the inability to persist in WT mice along with emergent lethality or virulence phenotypes in immunocompromised animals.

## VP1 F514I in CR6 is sufficient to confer lethality to *Stat1*⁻/⁻ mice after oral inoculation

Using a reverse genetics system for MNoV [29], we introduced a single point mutation (T to A) at viral nucleotide 6595 of the CR6 infectious molecular clone, thereby generating the VP1-F514I mutation in the background of an otherwise unmodified CR6 genome, designated CR6^F514I. We found that CR6^F514I replicated similarly to CR6 *in vitro* (**Figs 3A** and **S4A**). When administered to *Stat1*⁻/⁻ mice PO, CR6^F514I induced significant weight loss, as well as increased splenomegaly and lethality as compared to CR6 (**Fig 3B, 3C and 3D**). CR6^F514I had no effect on morbidity or mortality of WT mice. Intestinal infection of *Stat1*⁻/⁻ mice was similar between CR6 and CR6^F514I, with increased levels of viral shedding in the stool and enhanced viral levels in ileum and colon compared to WT animals at 5 dpi, with no observable differences associated with F514I (**Fig 3E and 3F**). In contrast, while *Stat1* deficiency permits enhanced systemic viral replication of CR6 [32,33] (**Fig 3F**), CR6^F514I genome copies were ~10 to 10,000-fold higher than CR6 genomes in *Stat1*⁻/⁻ MLN, spleen, and liver (**Fig 3F**). Importantly, only CR6^F514I infection of *Stat1*⁻/⁻ mice was detectable in the brain at 5 dpi (**Fig 3F**). While CR6^F514I exhibits enhanced viral spread and lethality in *Stat1*⁻/⁻ mice, of interest this virus replicated less effectively in the intestines of WT mice compared to CR6, with lower levels shed in the stool (**Fig 3E and 3F**). Thus, early in infection, F514I attenuates viral replication in the WT gut, but enhances systemic replication and lethality in mice with impaired IFN signaling.

## F514I does not prevent persistence of CR6

In order to determine whether F514I-mediated attenuation in the WT intestine was associated with a failure to persist, we analyzed CR6 and CR6^F514I infection in WT and *Stat1*⁻/⁻ mice to 21 dpi. Few *Stat1*⁻/⁻ mice infected with CR6^F514I survived past 7 dpi, permitting analysis of only a limited cohort; those that survived regained some weight, but remained underweight compared to other cohorts (**Fig 4A**). In WT mice at this persistent timepoint, intestinal viral genome levels were similar between CR6 and CR6^F514I challenged mice (**Fig 4B**). We analyzed whether CR6^F514I had reverted to the parental CR6 sequence via Sanger sequencing and found that I514 was maintained at 21 dpi (**S4B Fig**). While at early stages of infection, F514I results in reduced viral levels in intestinal tissues, at later stages, there appeared to be minimal impact of this mutation on viral loads. This observation is consistent with previous studies identifying a critical role for MNoV protein NS1, rather than VP1, in determining viral persistence in immunocompetent mice [27,37,49]. We also tested the effects of L514 on persistence by infecting WT mice with MNoV WU23. We found that this virus exhibited extraintestinal spread at acute timepoints similar to CW3, but was cleared from the spleen by 14 dpi, while it remained variably detectable at intestinal sites at persistent timepoints (**S5 Fig**). Together, these results indicate that I/L514 do not prevent persistence of MNoV.

## F514I spontaneously emerges in *Stat1*⁻/⁻ mice after oral inoculation

Examination of CR6-infected *Stat1*⁻/⁻ mice at 21 dpi revealed convergence of tissue viral levels to those found in the rare mice surviving CR6^F514I infection. Additionally, virus was detected in CR6-infected animals in the brain despite the lack of morbidity or mortality in this group (**Fig 4B**). We analyzed tissue samples from CR6-infected *Stat1*⁻/⁻ mice for emergence of the

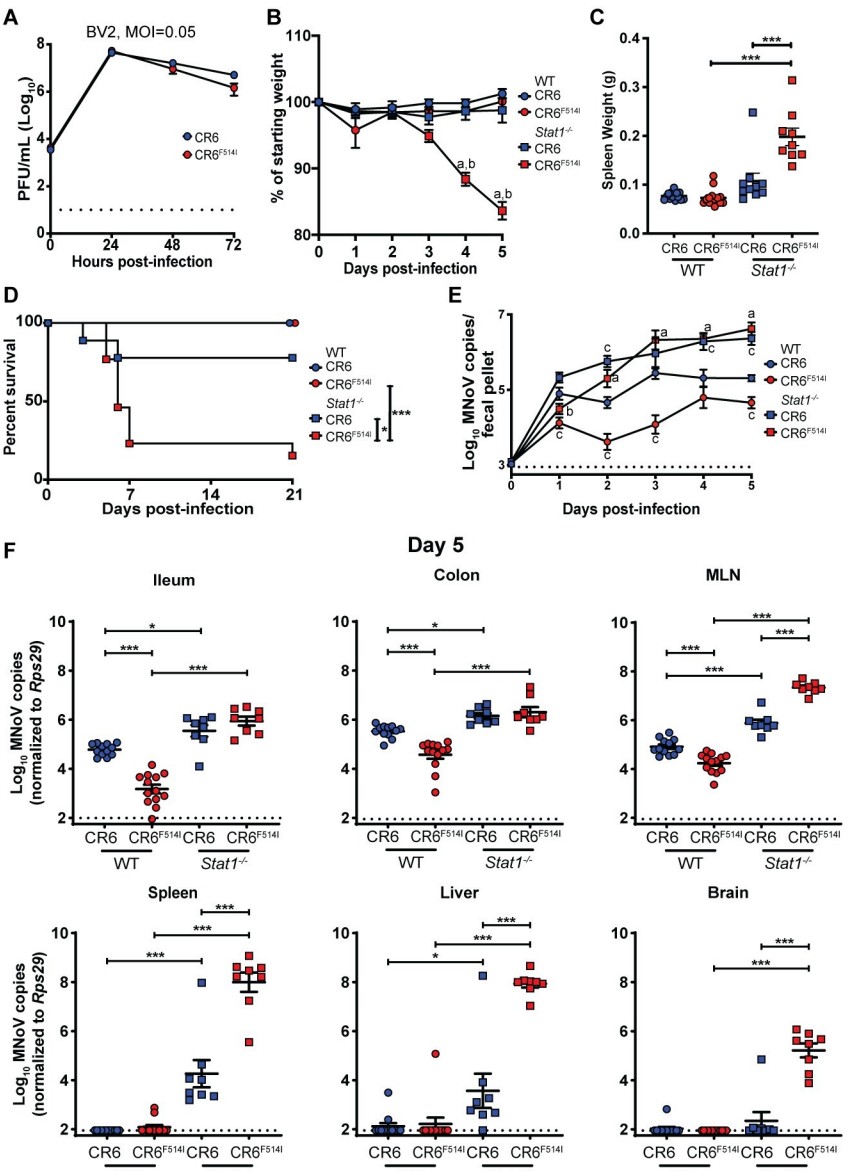

**Fig 3. VP1 F514I is sufficient to confer lethality to *Stat1*^-/- mice. (A)** Growth curves of CR6 and CR6^F514I in BV2 cells over 72 hpi, inoculated at an MOI of 0.05. Results combined from two independent experiments, analyzed by repeated-measures two-way ANOVA. **(B-F)** WT or *Stat1*^-/- mice were orally inoculated with 10^6 PFU of CR6 or CR6^F514I, and analyzed for weight loss to 5 dpi **(B)**, spleen weights at 5 dpi **(C)**, survival to 21 dpi **(D)**, stool viral levels to 5 dpi **(E)**, and tissue viral levels at 5 dpi **(F)**. N = 8–13 mice per group combined from at least two independent experiments. Weight loss **(B)** was analyzed by repeated-measures two-way ANOVA followed by Tukey's multiple-comparison test. Survival **(D)** was analyzed by log-rank (Mantel-Cox) test. Stool virus levels **(E)** were analyzed by mixed-effects analysis followed by Tukey's multiple-comparison test. Spleen weights **(C)** and tissue viral levels **(F)** were analyzed by ordinary two-way ANOVA followed by Tukey's multiple comparisons test. Dashed lines indicate limit of detection for assays. For panels **(B)** and **(E)**, letters indicate that the data point which they are next to have a statistically significant difference of $P < 0.05$ in comparison to: a) WT + CR6^F514I; b) *Stat1*^-/- + CR6; c) WT + CR6. Data shown as mean +/- SEM. ^***, $P < 0.001$; ^**, $P < 0.01$; ^*, $P < 0.05$; ns, not significant.

F514I amino acid change, and found that in one of three mice analyzed, the T6595A mutation (amino acid change F514I) was present in 21 dpi colon and liver tissues (**Fig 4C**). In light of this observation, we speculated that a single CR6-infected *Stat1*^-/- mouse sacrificed at day 5

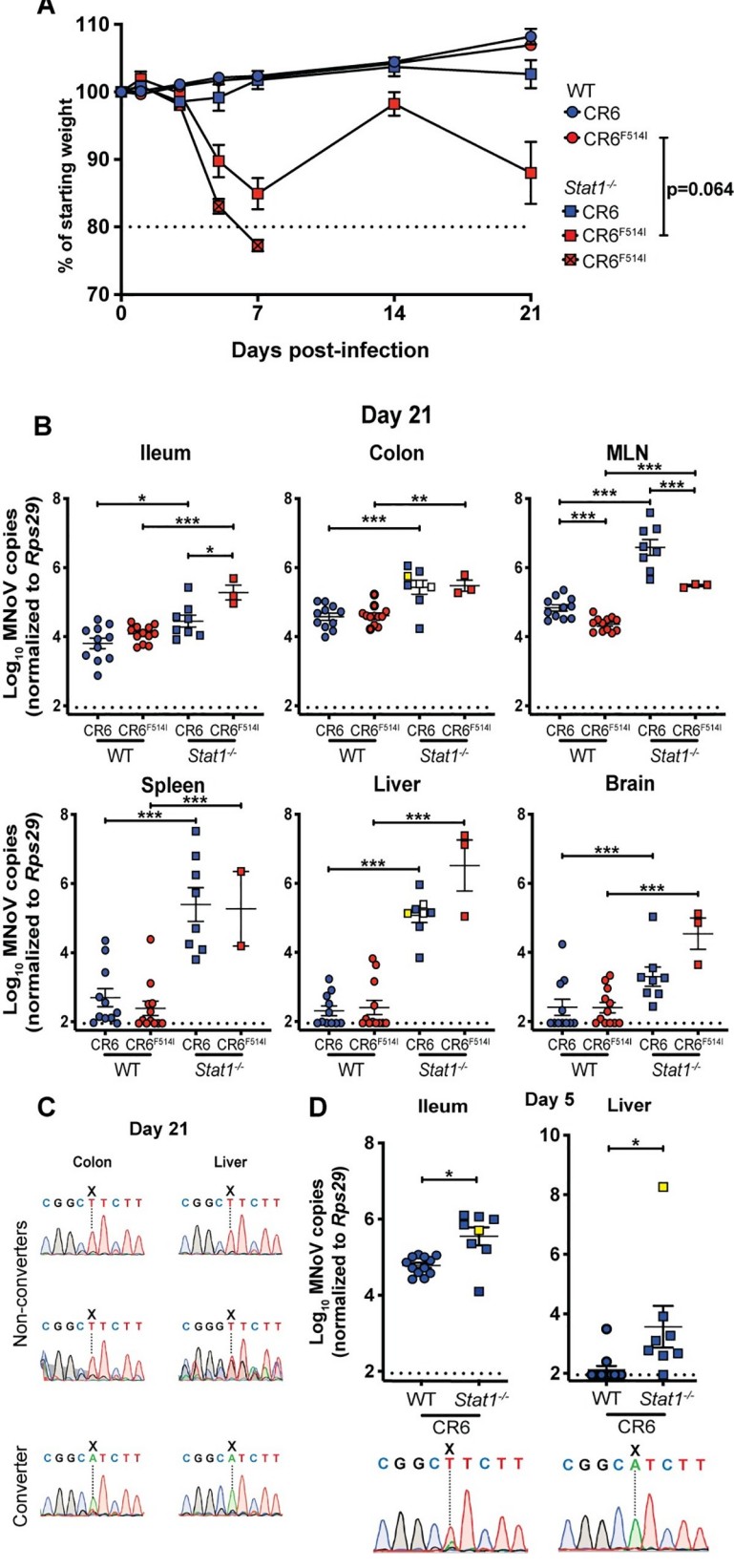

**Fig 4. VP1 F514I does not prevent persistence of CR6 and can emerge after PO inoculation in the absence of IFN signaling.** (A,B) WT or *Stat1*[-/-] mice were orally inoculated with $10^6$ PFU of CR6 or CR6[F514I], and analyzed for weight loss to 21 dpi (A), and tissue viral levels at 21 dpi (B). In (A), boxes with crosses indicate infected animals that died before day 7; dashed line indicates the 80% of initial weight cutoff at which mice were sacrificed. N = 9–13 mice per group over at least two independent experiments, with statistical analyses performed as in **Fig 3**. (C,D) cDNA from the indicated tissues (shown with yellow squares for the converter and white squares for non-converters) from individual *Stat1*[-/-] mice infected with CR6 and sacrificed at (C) 21 dpi or (D) 5 dpi (data repeated from **Fig 3F**) was PCR amplified and Sanger sequenced around nucleotide position 6595 of the MNoV genome; X indicates position 6595. Dashed lines indicate limit of detection for assays. ***, $P < 0.001$; **, $P < 0.01$; *, $P < 0.05$; ns, not significant.

with extraintestinal viral levels similar to CR6[F514I]-infected mice (**Fig 3F**) could be a rare case of F514I arising early after oral infection of immunocompromised mice. Indeed, F514I was the dominant variant in extraintestinal tissues but poorly-represented in intestinal tissue from this single outlier mouse (**Figs 4D and S6**). These findings suggest that F514I spontaneously emerges with moderate frequency via multiple different routes of inoculation (IC and PO) in IFN signaling-deficient mice.

## I514 is necessary for morbidity and extraintestinal spread in IFN-signaling deficient mice

Having observed that F514I was sufficient to confer lethality to mice lacking intact IFN signaling, we next tested whether I514 was required for CW3 to cause lethal infection. We introduced an A to T mutation at nucleotide 6595 in the CW3 plasmid to generate CW3[I514F] virus and confirmed that this virus replicated similarly to CW3 *in vitro* (**Fig 5A**). Similar to observations in *Stat1*[-/-] mice, CW3 causes rapid lethality in *Ifnar1*[-/-]*Ifngr1*[-/-] mice which lack both type I and II IFN signaling [50,51]. We infected WT and *Ifnar1*[-/-]*Ifngr1*[-/-] mice, due to mouse availability, with CW3 or CW3[I514F], and found that while CW3 infection led to dramatic weight loss of *Ifnar1*[-/-]*Ifngr1*[-/-] mice by 3 dpi, CW3[I514F] did not cause the same rapid morbidity (**Fig 5B**). Tissues were harvested at 3 dpi prior to a requirement for euthanasia due to excessive weight loss, and substantial differences in tissue viral levels were identified. Both intestinal and extraintestinal tissues from *Ifnar1*[-/-]*Ifngr1*[-/-] mice infected with CW3 exhibited significantly higher viral replication than those from WT mice (**Fig 5C**), with ~10,000-fold higher viral levels detected in MLN, spleen, liver, and brain. In contrast, CW3[I514F] infection of *Ifnar1*[-/-]*Ifngr1*[-/-] mice was highly attenuated, with only a ~100-fold increase in viral genomes in MLN and spleen compared to WT mice, and scant virus detected in the brain (**Fig 5C**). Additionally, we observed a modest increase in colonic viral loads of CW3[I514F] relative to CW3 in WT mice (**Fig 5C**), reminiscent of the increased intestinal loads of CR6 compared to CR6[F514I] (**Fig 3F**), further supporting a role for F514 in permitting intestinal MNoV replication. These data suggest that in addition to I514 being sufficient to confer morbidity and mortality (**Fig 3B and 3D**), this amino acid is also necessary to mediate morbidity and rapid viral spread to extraintestinal tissues, including the brain, in an IFN signaling-deficient background.

## I514- and L514-mediated lethality are independent of tuft cell infection

We next sought to determine how F514I confers increased lethality to MNoV in IFN-deficient mice. Since tuft cells are the major target cell of CR6 *in vivo* [30], we assessed whether tuft cells were necessary for F514I-associated lethality by infecting *Stat1*[-/-] mice with (*Pou2f3*[+/-]*Stat1*[-/-]) or without (*Pou2f3*[-/-]*Stat1*[-/-]) intestinal tuft cells. The *Pou2f3* gene has been shown to be essential for tuft cell development and *Pou2f3*[-/-] mice are tuft cell-deficient [52,53]. We observed that oral inoculation of these mice with CR6[F514I] led to equivalent lethality and splenomegaly between strains (**Fig 6A and 6B**), indicating that tuft cells are dispensable for CR6[F514I]-

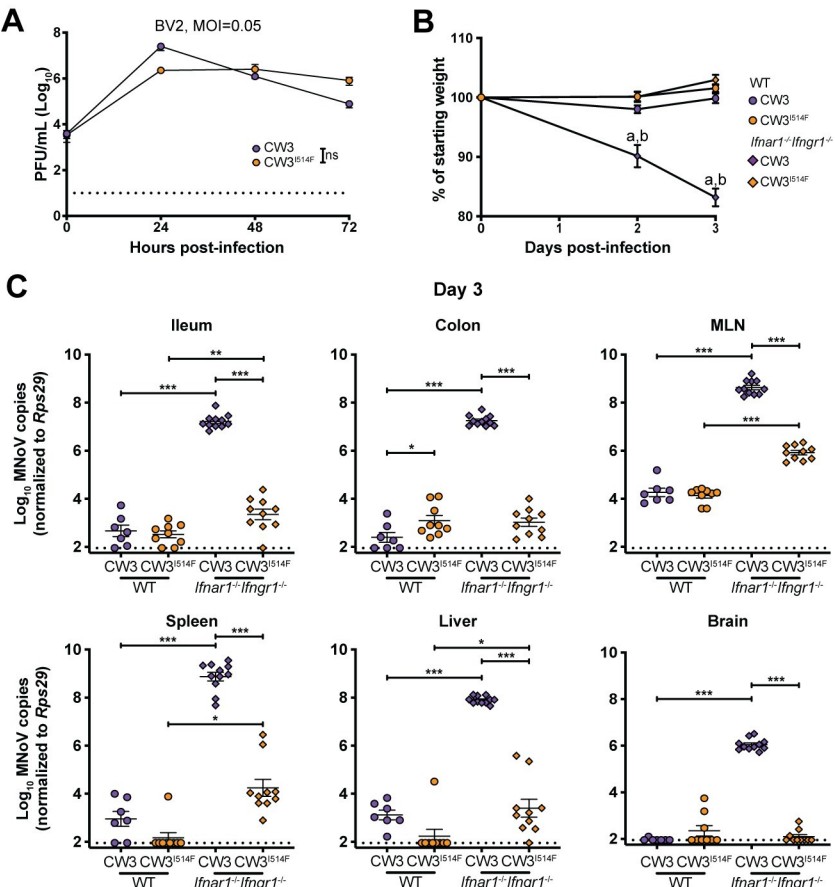

**Fig 5. VP1 I514 is necessary for early morbidity and brain replication in IFN signaling-deficient mice. (A)** Growth curves of CW3 and CW3$^{I514F}$ in BV2 cells over 72 hpi, inoculated at an MOI of 0.05. Results combined from two independent experiments. **(B,C)** WT or *Ifnar1$^{-/-}$Ifngr1$^{-/-}$* mice were orally inoculated with $10^6$ PFU of CW3 or CW3$^{I514F}$, and analyzed for weight loss to 3 dpi **(B)**, and tissue viral levels at 3 dpi **(C)**. N = 7–11 mice per group over two independent experiments with statistical analyses performed as in **Fig 3**. Dashed lines indicate limit of detection for assays. For panel **(B)**, letters indicate that the data point which they are next to have a statistically significant difference of *P* < 0.005 in comparison to: a) WT + CW3; b) *Ifnar1$^{-/-}$Ifngr1$^{-/-}$* + CW3$^{I514F}$. ***, *P* < 0.001; **, *P* < 0.01; *, *P* < 0.05; ns, not significant.

mediated mortality. Analysis of stool, ileum, colon, MLN, and spleen viral levels in survivors at 14 dpi demonstrated equivalent viral levels in *Pou2f3$^{+/-}$Stat1$^{-/-}$* and *Pou2f3$^{-/-}$Stat1$^{-/-}$* (**Fig 6C and 6D**), supporting that CR6$^{F514I}$ infection of intestinal and extraintestinal tissues of *Stat1$^{-/-}$* mice is tuft cell-independent. We additionally infected *Pou2f3$^{+/-}$Stat1$^{-/-}$* and *Pou2f3$^{-/-}$Stat1$^{-/-}$* with MNoV strain WU23 containing L514, and found that while lethality was delayed in *Pou2f3$^{-/-}$Stat1$^{-/-}$* mice compared to *Pou2f3$^{+/-}$Stat1$^{-/-}$* littermates, all mice succumbed to infection (**Fig 6E**). Thus, L514 is also associated with enhanced virulence, and both I514- and L514-associated lethality in *Stat1$^{-/-}$* mice does not require tuft cells.

## Disruption of *Stat1* in hematopoietic cells is sufficient for F514I-mediated extraintestinal spread

With the observation that tuft cells were not required for F514I-mediated lethality, we next sought to define the cell types in which *Stat1* expression was required to prevent systemic spread and morbidity. MNV-1 has been shown to infect macrophages, dendritic cells, B cells

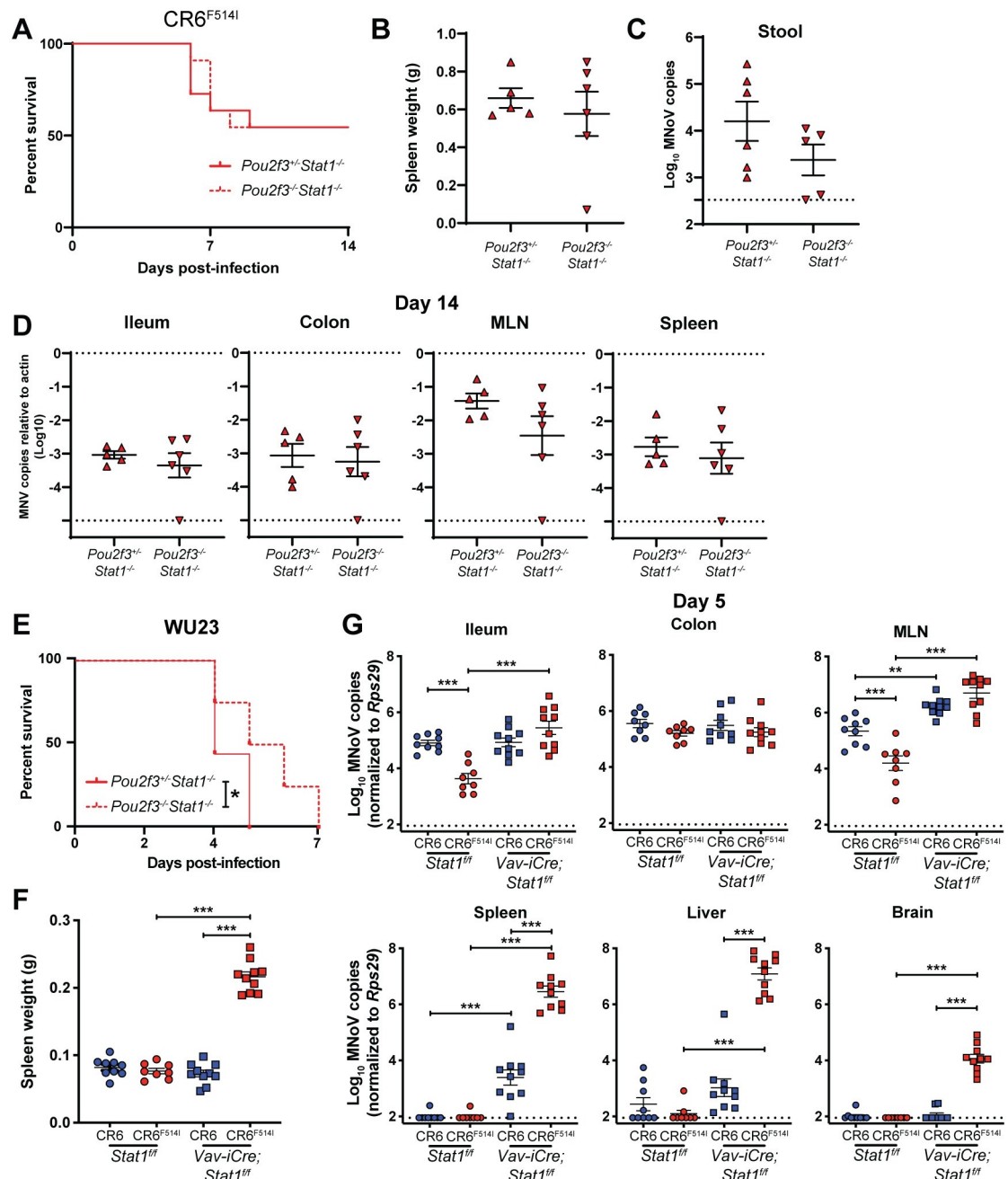

**Fig 6. Tuft cells are dispensable for, but *Stat1* disruption in hematopoietic cells permits, I514- or L514-associated pathogenesis. (A,B)** *Pou2f3*^+/−^*Stat1*^−/−^ and *Pou2f3*^−/−^*Stat1*^−/−^ were orally inoculated with $10^6$ PFU of CR6^F514I^ and analyzed for survival to 14 dpi (**A**), splenomegaly (**B**), stool viral levels (**C**), and tissue viral levels at 14 dpi (**D**). N = 11 mice per group over three independent experiments with 5–6 mice per group analyzed at 14 dpi. **(E)** *Pou2f3*^+/−^*Stat1*^−/−^ and *Pou2f3*^−/−^*Stat1*^−/−^ were orally inoculated with $10^6$ PFU of WU23 and analyzed for survival to 7 dpi. N = 8–9 mice per group over two independent experiments. **(F,G)** *Stat1*^f/f^ or *Stat1*^f/f^-Vav-iCre mice were orally inoculated with $10^6$ PFU of CR6 or CR6^F514I^ and analyzed for spleen weights **(F)** and tissue viral levels at 5 dpi **(G)**. N = 8–10 mice per group over two independent experiments, with statistical analyses performed as in **Fig 3**. ^***^, $P < 0.001$; ^**^, $P < 0.01$; ^*^, $P < 0.05$; ns, not significant.

and T cells in gut-associated lymphoid tissue in WT mice [36], and we thus infected *Stat1*^f/f^ mice crossed to Vav-iCre (expressed in all hematopoietic cells and tuft cells) with CR6 or CR6^F514I^ [54,55]. Ablation of *Stat1* in these cells was sufficient for CR6^F514I^ to confer

significant splenomegaly at 5 dpi (**Fig 6F**), as well as robust extraintestinal viral replication in spleen, liver, and brain (**Fig 6G**). In sum, these data support that ablating IFN-signaling in the hematopoietic lineage permits I514-mediated MNoV pathogenesis.

### I514 supports recruitment of inflammatory cells to Peyer's patches and extraintestinal MNoV spread

The CW3 capsid, in comparison to the CR6 capsid, stimulates recruitment of MNoV- susceptible inflammatory monocytes and neutrophils to intestinal Peyer's patches (PP) in *Stat1*$^{-/-}$ mice (38). We tested whether the F514I amino acid change was sufficient to confer these phenotypes to CR6. Consistent with this previous report, we observed increased inflammatory monocytes recruited to PP and MLNs in CW3-infected *Stat1*$^{-/-}$ mice compared to naïve and CR6-infected *Stat1*$^{-/-}$ mice (**Fig 7A**). While some *Stat1*$^{-/-}$ tissues, particularly the PPs, infected with CR6$^{F514I}$ had increased inflammatory monocyte recruitment at 2 dpi, the mean recruitment in both tissues remained less than that of CW3 infected tissues and the difference was not statistically significant. However, an increased proportion of monocytes infected by CW3 and CR6$^{F514I}$ were present in the MLNs at 2 dpi compared to CR6 (**Fig 7B**). As reported previously [38], there were high levels of variation between individual mice in these phenotypes, but in sum these data suggest that F514I may partially contribute to the inflammatory monocyte recruitment and infection phenotypes observed with the full CW3 capsid. Having observed decreased viral loads in the intestine at 5 dpi (**Fig 3F**), we additionally quantified viral loads in intestinal tissues and isolated Peyer's patches from WT animals at 1 dpi to examine early events in CR6$^{F514I}$ infection. Consistent with our 5 dpi data, there were significantly fewer viral genomes in the ileum and colon of mice infected with CR6$^{F514I}$ at 1 dpi as compared to CR6 (**Fig 7C**). However, in the Peyer's patches, we observed a trend toward higher levels of infection by CR6$^{F514I}$ than CR6, supporting that F514I may confer a bias during early infection away from intestinal tuft cells towards immune cell populations in Peyer's patches.

As our flow cytometric data suggested that F514I increases the ability of MNoV to infect immune cells, we sought to confirm this by testing whether CR6$^{F514I}$ was trafficked in the blood to facilitate dissemination to the brain. Unlike CR6, which was generally undetectable in the blood of WT and *Stat1*$^{-/-}$ mice, CR6$^{F514I}$ was present at high levels in the blood at 5 dpi and was substantially enriched in the cellular fraction of the blood (**Fig 7D**). As CW3 can infect lymphocytes *in vivo* and *in vitro* [36], we assessed the ability of CR6 and CR6$^{F514I}$ to infect the immortalized B cell line M12 (**S8 Fig**). Our data demonstrate that, unlike CW3, CR6$^{F514I}$ cannot replicate in B cells *in vitro*, suggesting the infected circulating blood cells are likely not lymphocytes. Thus, our data supports a model in which I514 confers enhanced recruitment and infection of immune cells in Peyer's patches and the MLN. I514 virus can subsequently disseminate rapidly to extraintestinal tissues via non-lymphoid immune cells in the blood in *Stat1*$^{-/-}$ mice, permitting lethal infection of the brain.

## Discussion

We sought to determine if acute, systemic MNoV strains could have originally evolved in immunocompromised mice from persistent, enteric strains present endemically in mouse colonies. Isolation of the first discovered MNoV strain, MNV-1, involved serial intracranial passaging of virus in mice lacking IFN signaling [23]; we sought to recreate this scenario by administering a persistent strain intracranially in *Stat1*$^{-/-}$ mice. Emergence of virus with lethal potential when administered orally occurred within one week. Deep sequencing of the viral genome revealed a consistent mutation primarily observed in intracranially-inoculated *Stat1*$^{-/-}$ mice: T6595A, which results in an F514I mutation in VP1, a polymorphism present in

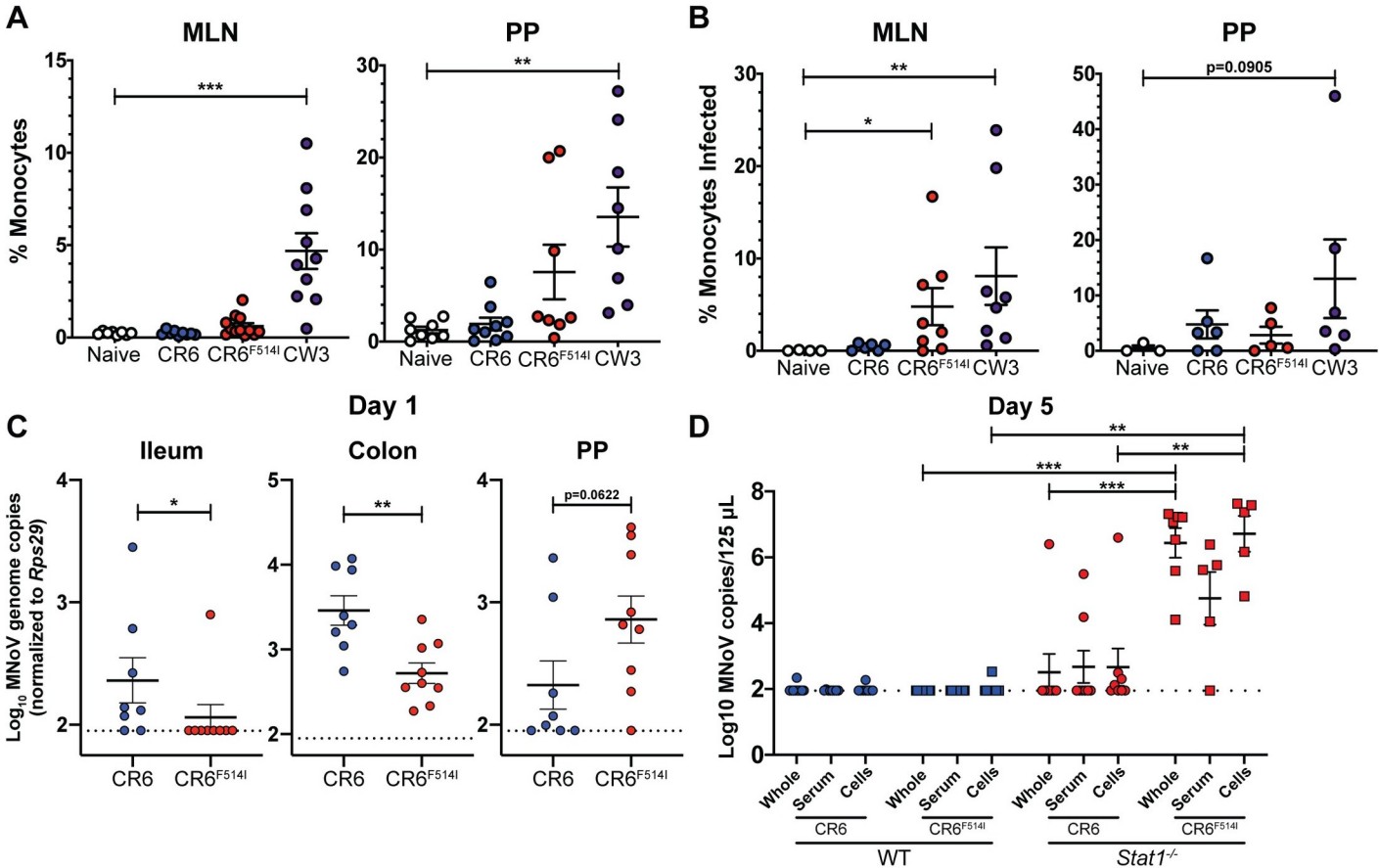

**Fig 7. I514 results in increased monocyte recruitment early in infection and preferentially infects lymphoid tissue within the gut, leading to extraintestinal spread via the blood. (A,B)** *Stat1*$^{-/-}$ mice were inoculated orally with $10^6$ PFU of the indicated viral strains or vehicle control. At 2 dpi, mice were sacrificed and single-cell suspensions of the MLNs and Peyer's patches (PP) were analyzed by flow cytometry for relative percentages of monocytes **(A)** and infected monocytes **(B),** with gating performed as shown in **S7 Fig**. N = 3–10 mice per group from two-five independent experiments. Data were analyzed by Kruskal-Wallis test followed by Dunn's multiple comparisons test using naïve mice as a comparator. **(C)** WT mice were inoculated orally with $10^6$ PFU of the indicated viral strains and sacrificed at 1 dpi. Ileum, colon, and Peyer's patches were harvested and MNoV genome copies were quantified via qPCR. N = 8–9 mice per group from two independent experiments. Groups were assessed for normality with the D'Agostino and Pearson test; tissues with normal distributions (Peyer's patches and colon) were compared with Welch's t-test, while tissues with non-normal distributions were compared with Mann-Whitney test. **(D)** WT and *Stat1*$^{-/-}$ mice were inoculated orally with $10^6$ PFU of the indicated viral strains and blood was harvested at 5 dpi via submandibular bleed. RNA was extracted from whole blood, as well as separated cellular and serum blood fractions, and MNoV genome copies were quantified via qPCR. Groups were compared via a mixed-effect analysis with the Geisser-Greenhouse correction, followed by Tukey's multiple comparisons test. N = 5–12 mice per group from at least two independent experiments. $^{***}$, $P < 0.001$; $^{**}$, $P < 0.01$; $^{*}$, $P < 0.05$; ns, not significant.

genomes of acute strains MNV-1 and the derived CW3 clone. F514I confers the capacity for MNoV to rapidly move systemically, including to the brain, thereby causing lethality in the majority of *Stat1*$^{-/-}$ mice. We found this single amino acid could convert CR6 into a lethal virus and, when removed, rendered CW3 significantly less pathogenic. Thus, F514I is sufficient and necessary for MNoV-mediated pathogenesis in IFN signaling-deficient mice.

Of interest, this mutation also appears to spontaneously emerge during intrahost evolution in a subset of *Stat1*$^{-/-}$ mice infected orally. We speculate that the mutation spontaneously evolves from CR6 only when the associated delay in systemic dissemination is sufficient to permit adaptive immune responses to develop. These adaptive responses would then protect infected animals from the morbidity and mortality associated with CR6$^{F514I}$ infection. Only when *Stat1* is absent and systemic dissemination occurs more rapidly can this mutation appear before being controlled by adaptive immunity. Intestinal CR6 infection can elicit both

humoral and cellular adaptive immune responses, but these are insufficient to clear intestinal virus [31,56]. However, previous studies have shown that high systemic MNoV viral loads can be associated with robust functional adaptive responses even when type I IFN signaling is partially abrogated [57], supporting the possibility that adaptive responses may protect *Stat1*[-/-] mice from systemic infection if given time to develop. Thus, by accelerating the speed and amplifying the frequency with which F514I emerged, as well as intensifying the observed phenotypic effect, intracranial inoculation appears to have helped expose a mutational predilection of MNoV.

CR6 has exclusively been detected in tuft cells during infection [30], while MNV-1 and CW3 acutely infect immune cells in Peyer's patches and the MLN [36]. The capsid (VP1) of CW3, specifically the protruding (P) domain and not the shell domain, stimulates inflammatory cytokines and IFNs thereby driving recruitment of inflammatory monocytes and neutrophils which are subsequently infected [38]. The P domain has also been clearly implicated as being necessary and sufficient for increased extraintestinal replication and virulence in *Stat1*[-/-] mice [29]. It has previously been reported that CR6 with the CW3 P1 subdomain does not exhibit additional virulence, and that CW3 with the P2 domain of CR6 is attenuated, both observations suggesting that the P2 subdomain of CW3 is a mediator of virulence [29]. Further, the apical P2 subdomain directly binds CD300LF, the MNoV receptor [43–45]. 514 is a residue in the P1 subdomain of the P domain, which is located in closer proximity to the capsid shell domain (**Fig 8A and 8B**). This residue is not surface exposed, and structures of CW3

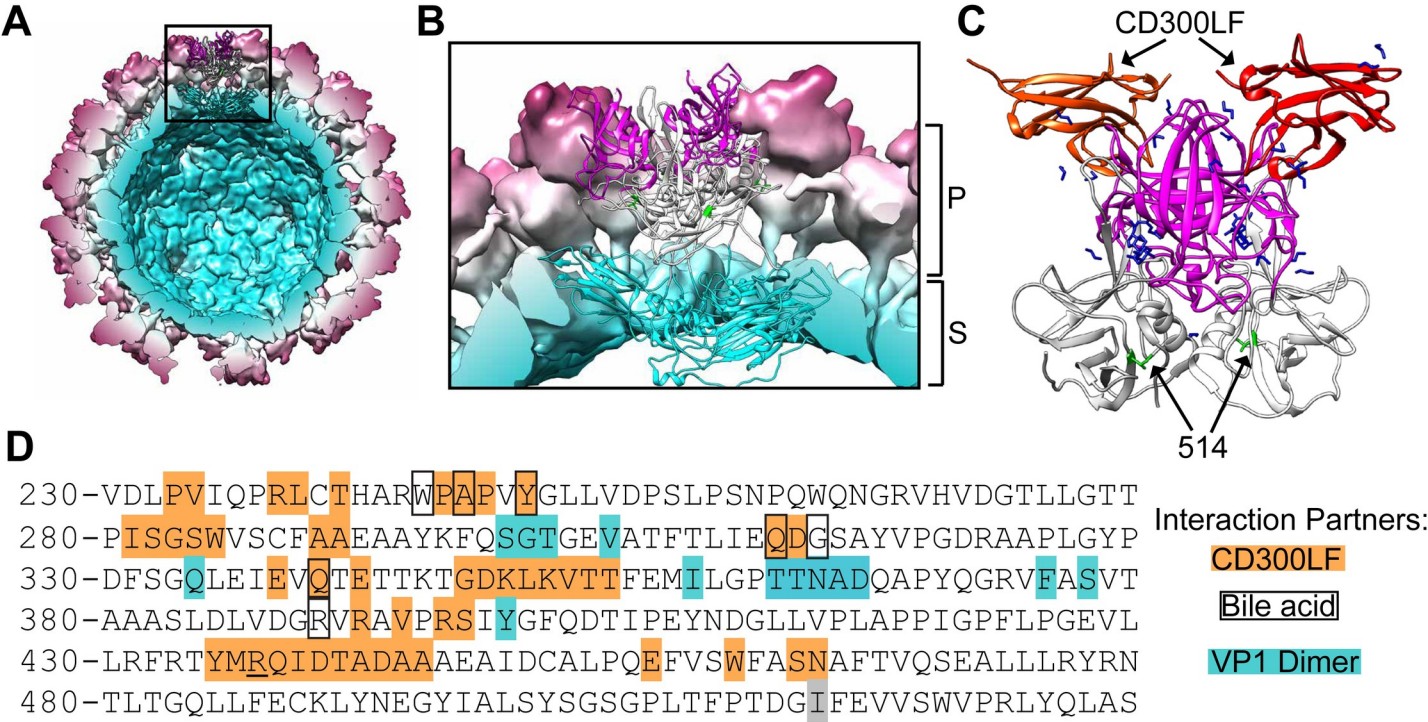

**Fig 8. VP1 514 is a deeply-buried, non-surface exposed residue within the MNoV capsid. (A)** A reconstruction of the CW3 capsid (PDB: 6CRJ), with the shell, P1, and P2 domains (residues 1–225; 226–277 and 416–540; 278–415, respectively, as defined previously (43)) in cyan, white, and magenta, respectively. Amino acid 514 is highlighted in green. **(B)** An expanded view of a single asymmetric trimer of VP1 within the CW3 capsid from the inset in **(A)**, colored as in **(A)** with the P and S domains of VP1 bracketed. **(C)** A 3-dimensional structure of the CW3 VP1 dimer (P1 and P2 domains only) in complex with two copies of the MNoV receptor CD300LF (PDB: 6E47). Colored as in **(A)**, with CD300LF additionally colored in orange/red and ligands (including bile acid glycochenodeoxycholic acid) in dark blue. **(D)** Amino acid sequence of CW3 VP1 residues 230–529, with known residues involved in intermolecular interactions highlighted. The grey highlight indicates residue 514, orange highlights indicate residues that interact with CD300LF, black boxes indicate residues that interact with bile acids, and cyan highlights indicate residues involved in interactions with the other copy of VP1 within the dimer. Adapted from (43). All images created using UCSF Chimera (63).

VP1 in complex with MNoV receptor CD300LF, bile acids, and divalent cations indicate that residue 514 is not involved in interactions with the receptor, known cofactors, or other copies of VP1 within the capsid (**Fig 8C and 8D**). Despite this, the identity of this specific amino acid mediates the recruitment of inflammatory cells to infect and facilitate extraintestinal spread. This observation is surprising in the context of the previous literature, which would have better supported a role for specific P2 amino acids in mediating altered tropism and increased virulence. However, recent reports have suggested that norovirus capsid structures may be highly dynamic and that the P1 domain may govern this flexibility by regulating interactions with the shell domain [58–62]. We hypothesize that residue 514 may be involved in this flexibility, as several amino acids with which it is predicted to interact make up a portion of this flexible linker, in the capsids of both MNoV S7 (which possesses F514) and CW3 (which possesses I514) (**S9 Fig**). F/I514 may influence this linker to adopt either the "expanded" or "closed" conformation, changing the capsid structure and altering its ability to interact with receptors and/or the immune system. However, this remains to be experimentally demonstrated and further experiments and structural analyses will need to be conducted to explore this possibility.

We recently reported another surprising context in which F514I emerges: after intraperitoneal inoculation of $Cd300lf^{+/-}Stat1^{-/-}$ and $Cd300lf^{-/-}Stat1^{-/-}$ mice with high-dose CR6 [63]. Compared to lethal infection of $Cd300lf^{+/-}Stat1^{-/-}$ littermates, intraperitoneal CR6 infection of mice lacking MNoV receptor CD300LF is significantly abrogated and is non-lethal. However, virus, specifically CR6 with the F514I mutation, is detectable in extraintestinal tissues from both $Cd300lf^{+/-}Stat1^{-/-}$ and $Cd300lf^{-/-}Stat1^{-/-}$ mice. Our current work indicates that F514I stimulates recruitment and infection of inflammatory monocytes in Peyer's patches. Exploration of whether this particular cell type is permissive to infection by MNoV via alternate receptor CD300LD [44,45], is of future interest.

It has previously been reported that the CW3 capsid stimulates greater levels of type I and III IFNs in Peyer's patches [32,38], associated with impaired replicative fitness in the intestine. Accordingly, CR6$^{F514I}$ growth in the intestine is significantly abrogated at acute timepoints in IFN signaling-intact mice, consistent with higher levels of IFNs limiting intestinal MNoV infection [32,64–66]. This likely explains the predominance of F514 among known MNoV strains, with I/L514 being observed solely in uniquely virulent strains isolated from immunocompromised animals (**Fig 2D**). As MNoV is primarily spread via the fecal-oral route, a virus shed at higher levels upon infection has substantial advantages in persisting within mouse colonies. Even if the F514I mutation were to arise within the viral quasispecies in a WT mouse, this variant would be quickly selected against during transmission, preventing this and related virulent mutant strains from coming to dominate within a mouse colony. We speculate that it is only when IFN signaling is ablated (as was the case for the mice from which MNV-1 and WU23 were isolated) that viruses with mutations such as F514I, which allow for greater extraintestinal spread and lethality, can persist, as they are shed similarly to avirulent strains in IFN signaling-deficient animals. In the absence of competition from more shedding-competent variants, however, CR6$^{F514I}$ maintains the capacity to persist and recovers to levels similar to CR6 at persistent timepoints. MNoV persistence is VP1-independent and has instead been mapped to the secreted viral protein NS1, wherein a glutamic acid at amino acid 94 (E94) is sufficient for viral persistence [27,37,49]. The retained persistence of CR6$^{F514I}$ and WU23 is thus likely secondary to the presence of E94 in the NS1 protein of both viruses.

In the absence of IFN signaling pressure, I514 confers an enhanced ability to replicate systemically, explaining its emergence as the dominant sequence in a variety of $Stat1^{-/-}$ contexts. We propose that emergence of I514 provides one key evolutionary step behind the emergence of acute pathogenic MNoVs in immunocompromised mice. The fortuitous initial discovery of an acute pathogenic MNoV strain, followed by subsequent identification of a variety of

persistent non-pathogenic strains, has provided a powerful set of comparison strains to reveal key insights into enteric RNA virus infection and regulation [23,28]. Similar to the many key immunological comparisons made using the systemic RNA virus lymphocytic choriomeningitis virus strains Armstrong and Clone 13 [67], study of acute versus persistent strains of MNoV has informed understanding of innate and adaptive immune responses to enteric viruses, microbiota-virus interactions, tolerance to oral antigens, and mechanisms governing cellular tropism and persistence [27,30–33,37,56,68]. Additional discoveries from use of this critical comparison system are doubtless forthcoming.

Detailed exploration of the mutations that emerge in viral genomes under different *in vitro* and *in vivo* conditions has proven immensely useful to identify viral residues that interact with or antagonize host factors for many clinically important viruses. The application of this general approach to norovirus has been limited, but use of the simple and effective method for enriching viral genomes prior to sequencing described here should make viral evolutionary studies in MNoV and HNoV systems more tractable. Recent studies have indicated that distinct HNoV variants emerge in immunocompromised hosts, and indeed that this chronically-shed virus maintains infectivity [48,69–71]. Careful tracking of viral genomic characteristics during chronic HNoV infection has the potential to facilitate identification of novel variants emerging in immunocompromised settings that could seed future epidemics. Evolutionary analysis permits novel insights into both the origins of microbial characteristics, as described here, as well as into specific putative viral vaccine and therapeutic targets, to be explored in future studies.

## Materials & methods

### Ethics statement

All experiments at Washington University and Yale University were conducted according to regulations stipulated by the Washington University or Yale University Institutional Animal Care and Use Committees and to animal protocols 20140244 and 20190126, approved by the Washington University Animal Studies Committee, or 2018–20198, approved by the Yale University Office of Animal Research Support.

### Mouse lines

C57BL/6J wild-type (WT) mice were originally purchased from Jackson Laboratories (stock #000664, Jackson Laboratories, Bar Harbor, ME) and bred and housed in Washington University in Saint Louis animal facilities under specific pathogen free, including murine norovirus-free, conditions. $Stat1^{-/-}$ mice (B6.129S(Cg)-$Stat1^{tm1Dlv}$/J) [72] were maintained in the same conditions. $Stat1^{f/f}$ (B6;129S-$Stat1^{tm1Mam}$/Mmjax) were crossed to Vav-iCre (B6.Cg-$Commd10^{Tg(Vav1-icre)A2Kio}$/J) and Cre- and Cre+ littermates were used in experiments. $Ifnar1^{-/--}$ $Ifngr1^{-/-}$ were generated from crossing of $Ifnar1^{-/-}$ [73] and $Ifngr1^{-/-}$ [74] lines.

$Pou2f3^{-/-}$ mice were developed with assistance from the Genome Engineering and iPSC Center (GEiC) at Washington University in Saint Louis, where gRNA 5'-AGGCCATGC-CACCTGAGCCANGG-3' was designed to target the $Pou2f3$ locus in the fourth exon. C57BL/6J (Jackson Laboratories, Bar Harbor, ME) fertilized zygotes were injected with Cas9 mRNA and gRNA. A founder mouse with the following mutation was recovered:

WT CCCACAGGCCATGCCACCTGAGCCAAGGAC
KO CCCACAGGCCT-—-—-—-—CTCCCAAGGAC
* * * * * * * * * * * * * * * * * *

Additional generations were genotyped by Transnetyx (Cordova, TN) from tail biopsy specimens using real-time PCR with mutation-specific probes. $Pou2f3^{-/-}$ mice were crossed to $Stat1^{-/-}$ mice, and littermate $Pou2f3^{+/-}Stat1^{-/-}$ and $Pou2f3^{-/-}Stat1^{-/-}$ mice were used in

experiments conducted at Yale University. A trio breeding scheme was used for mice to support use of littermate controls for experiments.

## Generation of viral stocks

Stocks of MNoV strains CR6 and CW3 were generated from molecular clones as previously described [29]. Briefly, plasmids encoding the viral genomes were transfected into 293T cells to generate infectious virus, which was subsequently passaged on BV2 cells. After two passages, BV2 cultures were frozen and thawed to liberate virions. Cultures then were cleared of cellular debris and virus was concentrated by ultracentrifugation through a 30% sucrose cushion. Titers of virus stocks were determined by plaque assay on BV2 cells [44].

The T6595A mutation was introduced into plasmid pCR6 using inverse PCR. PCR was performed using primers FW032 (5'-GACGGCATCTTTGAGGTTGTCAG-3') and FW033 (5'-AGTGGGGAAAGTGAGGGGG-3'), using pCR6 as a template. The PCR product was run on an agarose gel, and the appropriate band was excised and purified using the Monarch Gel Extraction Kit (NEB). The purified product was treated with DpnI (NEB) to remove template DNA, then phosphorylated with T4 polynucleotide kinase (NEB) and ligated with T4 DNA Ligase (NEB). The circularized plasmid was then transformed into chemically competent *E. coli* DH5α, and transformants were selected on LB agar plates with 100 μg/mL ampicillin. Individual colonies were picked and propagated, and plasmid was purified using the QIAprep Spin Miniprep Kit (Qiagen). Successful introduction of the intended mutation was confirmed via Sanger sequencing using primer FW037 (5'-GGAGTTTATCTCCTGGTTTGCAAGC-3').

The A6595T mutation was introduced into plasmid pCW3 using site-directed mutagenesis [75]. Amplification was performed as described [75] using primers FW103 (5'-CCGACCGATGGCTTCTTTGAGGTCGTCAG-3') and FW104 (5'-CTGACGACCTCAAAGAAGCCATCGGTCGG-3') with Phusion Polymerase (NEB). The product was purified with the EZNA Cycle Pure Kit (Omega Bio-tek) and digested with DpnI. Digested product was transformed and purified as above and the mutation was confirmed via Sanger sequencing with primer FW069 (5'-CAGGCCCCCTACCAGGG-3'). Mutant viruses were propagated as for CR6 and CW3 above, without concentration through a sucrose cushion.

## Murine norovirus infections

For peroral MNoV infections, mice were inoculated with a dose of $10^6$ PFU of the indicated strain orally in a volume of 25 μl or via gavage in a volume of 200 μl. For brain homogenate infection, a 1:80 dilution of 2-ml brain homogenate was used for volumetric matching to our standard dose of 25 μl. For intracranial infections, mice were anesthetized by intraperitoneal administration of ketamine/xylazine, then inoculated with a dose of $10^6$ PFU of CR6 in a volume of 10 μl. Mice were monitored until they regained consciousness for any adverse effects of inoculation.

Stool and tissues were collected between one and thirty days post-infection. All stool and tissues were harvested into 2-ml tubes (Sarstedt, Germany) with 1-mm-diameter zirconia/silica beads (Biospec, Bartlesville, OK). Tissues were flash frozen in a bath of ethanol and dry ice and either processed on the same day or stored at −80°C.

## *In vitro* murine norovirus experiments

MNoV growth curves and plaque assays were performed as described previously [44], with BV2 cells being frozen at 0, 24, 48, and 72 hours post-infection with MOI 0.05 of MNoV. Additionally, for plaque assays, $2\times10^6$ BV2s were seeded per well of a six-well plate. For growth curves in RAW264.7 cells, following infection with MOI 0.05 of MNoV supernatant fluid was

frozen at 0, 24, 48, 72, and 96 hours post-infection TCID$_{50}$ assays were performed using RAW264.7 cells and read at 7 dpi as described previously [76].

## RNA extraction and quantitative reverse transcription-PCR

As previously described [65], RNA was isolated from stool using a ZR-96 Viral RNA kit (Zymo Research, Irvine, CA). RNA from tissues or cells was isolated using TRI Reagent with a Direct-zol-96 RNA kit (Zymo Research, Irvine, CA) according to the manufacturer's protocol. For RNA extraction from blood, 250 μl of blood was harvested via submandibular bleed into EDTA-coated tubes. 125 μl was aliquoted as "whole" blood. The other 125 μl was transferred into a 1.5 mL microcentrifuge tube and centrifuged at 10,000xg for 5 minutes at 4˚C. The supernatant was removed and served as "serum". The pellet, serving as "cells", was washed once in 1 volume of DPBS and then resuspended in 1 volume of DPBS for RNA extraction. Following this separation, RNA was extracted as described above using the Direct-zol-96 RNA kit. 5 μl of RNA from stool or tissue was used for cDNA synthesis with the ImPromII reverse transcriptase system (Promega, Madison, WI). MNoV TaqMan assays were performed, using a standard curve for determination of absolute viral genome copies, as described previously [77]. Quantitative PCR for housekeeping genes *Rps29* or *Actin* was used to normalize absolute values of MNoV as previously described [33,63]. All samples were analyzed with technical duplicates.

## Sequencing of the murine norovirus genome

cDNA was prepared from total RNA isolated from fecal material as described above. The cDNA then underwent tagmentation using the Nextera DNA Library Preparation kit (Illumina) as previously described [78]. This was followed by PCR-mediated adapter ligation using 2x KAPA HiFi PCR master mix (Roche). The tagmented and indexed cDNA library was selected for approximately 200 base pair (bp) size using AMPure XP magnetic beads.

The Nextera DNA library was enriched using IDT xGen Lockdown hybridization capture system (IDT # 1072281) following the manufacturer's recommendation. 500ng of the Nextera DNA library was dried in a vacuum concentrator with 5 μg of Cot-1 DNA and 2ul of xGen NXT Universal Blockers (IDT #1079584). After hybridization buffer was added and the DNA incubated at 95˚C for 10 minutes, 4 μl of the xGen Lockdown probe pool was added and incubated at 65˚C for 4 hours. The custom MNoV probe pool consisted of an equal ratio of probes to both CR6 and CW3 strains. 120bp probes that spanned the full length of both ~7.5kb genomes were synthesized by IDT with the aid of their online tool. After incubation, the library-probe mixture was bound to streptavidin beads and extensively washed. The captured Nextera library was eluted in water, then amplified 14 cycles using 2x KAPA HiFi PCR and Illumina P5 and P7 primers (IDT # 1077675) then AMPure XP magnetic beads were used for post-capture PCR purification according to xGen Lockdown protocol. Equimolar quantities of libraries were run on the Illumina NextSeq platform using a paired-end 2 × 150 protocol.

Sequencing of the MNoV capsid was performed by PCR amplification from viral cDNA using primers 5'-CAACAACTTCACGGTCCAGTCGG3' and 5'-GCTTGAAAGAGTTGG CTTGGAGC-3' followed by Sanger sequencing by GENWIZ (South Plainfield, NJ) using the same primers.

## Analysis and visualization of murine norovirus sequencing and structure data

Raw sequencing reads were quality controlled with the BBTools suite [79] and aligned to the CR6 genome with Bowtie2 [80]. Variants were identified with the BCFtools "call" command

[81]. Additional annotation of called variants was performed with a custom R script (http://github.com/RachelRodgers/VirusVariantViewR). Visualization and inspection of alignment coverage and mutations was performed within a custom R Shiny application. Sequencing data have been uploaded to the European Nucleotide Archive with accession number PRJEB38177 (ERP121566).

Alignments of protein sequences were performed using Clustal Omega [82]. All visualizations of protein structures were performed using UCSF Chimera [83].

### Flow cytometry

Mesenteric lymph nodes and Peyer's patches were collected two days post-infection with the indicated viruses and forced through 70 μM cell strainers to create single cell suspensions. Cells were stained with LIVE/DEAD Aqua Stain (Invitrogen) in PBS according to manufacturer's instructions. Surface antigens were stained with the following antibodies: CD11b (clone M1/70, BioLegend Cat# 101216) and Ly6C (clone HK1.4, BioLegend Cat# 128025) and for blocking with Rat anti-Mouse CD16/CD32 (Mouse BD Fc Block, Clone 2.4G2, BD Cat# 553142). Cells were then stained for intracellular viral proteins; they were fixed and permeabilized using Cytofix/Cytoperm (BD) at room temperature for 10 minutes then washed in Perm/Wash buffer (BD) and incubated with 1:500 rabbit anti-NS1/2 or mouse anti-NS1 in Perm/Wash buffer for 1 hour at room temperature. Cells were then washed again and stained with secondary goat anti-rabbit antibody (Invitrogen, Cat# 550589) at a dilution of 1:500 in Perm/Wash buffer for 1 hour at room temperature. Data was collected on an LSR Fortessa (BD biosciences) and analyzed using FlowJo (TreeStar). Inflammatory monocytes were gated as $CD11b^+/Ly6C^{high}$.

### Statistical analysis

Data were analyzed with Prism 7 software (GraphPad Software, San Diego, CA). Mean +/- standard error of the mean are plotted in all graphs containing error bars. In all graphs, three asterisks indicate a $P$ value of $<0.001$, two asterisks indicate a $P$ value of $<0.01$, one asterisk indicates a $P$ value of $<0.05$, and ns indicates not significant ($P > 0.05$) as determined by Mann-Whitney test, one-way analysis of variance (ANOVA) or Kruskal-Wallis test, or two-way ANOVA with Tukey's multiple-comparison test, as specified in the relevant figure legends.

### Supporting information

**S1 Fig. Murine norovirus strain CR6 disseminates systemically by 3 dpi after intracranial inoculation in *Stat1*[-/-] mice. (A)** Schematics for control groups including intracranial (IC) passaging protocol in WT mice as well as per oral (PO) passaging protocols in WT and *Stat1*[-/-] mice, all initiated with $10^6$ PFU CR6 in three parallel "sets" of mice. Sacrifice of mice in these control groups was matched with spontaneous time of death of *Stat1*[-/-] IC experimental groups. For PO passaging, fecal material collected from mice at time of sacrifice was administered orally to the next round of mice. **(B,C)** Heatmap depicting murine norovirus levels detected by quantitative RT-PCR (qPCR) in designated tissues **(B)** or stool **(C)** at 3 dpi, with values ranging from $\log_{10}$ 1.95 to 9.50 (limit of detection at 2.0) for tissues and $\log_{10}$ 2.95 to 6.95 (limit of detection at 3.0) for stool. Each square indicates a single mouse tissue from 24 total mice analyzed.
(TIF)

**S2 Fig. Murine norovirus (MNoV) hybridization capture yields improved genomic sequencing efficiency, dependent upon initial viral levels. (A)** Schematic of enrichment protocol, in which cDNA from stool RNA undergoes Nextera tagmentation, followed by hybridization capture using biotinylated MNoV-specific probes and streptavidin beads (yellow hexagons) for pulldown prior to Illumina sequencing. **(B)** Stool samples collected from various mice infected PO with CR6 underwent this protocol, then pre- and post-enrichment samples were sequenced to assess efficacy of enrichment. **(C)** For stool samples from this study (**Figs 1D and S1C**) that were deep-sequenced after enrichment, the percentage of reads mapping to MNoV were compared to viral levels detected in samples by qPCR. Linear regression analysis for post-enrichment samples **(B)** and samples for this study **(C)** was performed using Graph-Pad Prism 7 software, with the P-value reporting if the slope is significantly non-zero.
(TIF)

**S3 Fig. F514I can appear rapidly after IC infection of *Stat1*<sup>-/-</sup> mice. (A)** Relative MNoV genomic coverage for those samples for which sequencing libraries could be successfully generated is shown. Range of genome coverage (with a cut-off of 100 reads per base) is shown to the right of each group in brackets. **(B, C)** Heatmap of Sanger sequencing-based analysis of position 6595 of stool **(B)** and indicated tissues **(C)** at 3 dpi as depicted in **S1B and S1C Fig**, with resulting encoded amino acid depicted by color. **(D)** Summary of mutations in VP1 found in MNoV CW3 and the most closely-related MNoV strains CR3, CR6, CR7 and WU23. Amino acids are sorted by VP1 domain, with the consensus amino acid (defined as being present at that position in >50% of the MNoV strains listed in **Fig 2D**) indicated. The amino acid at that position in CW3, CR3, CR6, CR7, and WU23 is listed if it differs from the consensus. Less closely-related MNoV isolates listed in **Fig 2D** were also aligned, but are not shown here as they do not uniquely possess any deviations from the consensus which are shared by CW3. Bolded text and * indicate variants that are unique to CW3.
(TIF)

**S4 Fig. F514I in CR6 is maintained at persistent timepoints and does not revert. (A)** Growth curves of CW3, CR6 and CR6<sup>F514I</sup> in RAW264.7 cells over 4 days, inoculated at an MOI of 0.05 Results combined from three independent experiments, analyzed by repeated-measures two-way ANOVA. **(B)** Sanger sequencing analysis of 21 dpi tissues from three independent WT mice infected with CR6<sup>F514I,</sup> with analyzed tissues depicted in yellow (data repeated from **Fig 4B**). One representative Sanger sequencing trace is depicted below the graph, with position 6595 indicated with *. Combined data from the three WT mice is shown by averaging the relative peak area for each base at position 6595 from Sanger sequencing traces.
(TIF)

**S5 Fig. Strain WU23, containing L514, exhibits acute extraintestinal infection and variable intestinal persistence.** WT mice were orally inoculated with $10^6$ PFU of MNoV WU23. Mice were sacrificed at 7 or 14 dpi, and the indicated organs were harvested and MNoV genome copies were quantified by qPCR. N = 5 mice per group from one experiment, with groups compared by Mann-Whitney test. Data shown as mean +/- SEM. **, $P < 0.01$; ns, not significant.
(TIF)

**S6 Fig. F514I spontaneously emerges in extraintestinal tissues in rare IFN signaling-deficient mice.** Sanger sequencing analysis of observed 5 dpi single "outlier" from *Stat1*<sup>-/-</sup> PO infection with CR6 (data repeated from **Fig 3F**). qPCR values from the "outlier" mouse are depicted in yellow, and Sanger sequencing traces for the area around nucleotide 6595 from each tissue are depicted beneath the graph for that tissue, with nucleotide 6595 indicated with

X. A relative quantification of the abundance of each base from the Sanger sequencing traces is also provided.
(TIF)

**S7 Fig. Gating strategy for MLN and Peyer's patches to identify monocytes and infected monocyte subsets.** Cells were gated as shown with a representative MLN sample from a *Stat1⁻ᐟ⁻* mouse infected with CW3. Cells were gated for live, single cells on the basis of FSC and SSC, followed by live/dead staining with LIVE/DEAD Aqua Stain. Inflammatory monocytes (**Fig 7A**) were identified as being CD11b⁺Ly6C⁺, and infected monocytes (**Fig 7B**) were identified as those inflammatory monocytes which were positive for MNoV NS1 or NS1/2.
(TIF)

**S8 Fig. F514I does not enhance replication of CR6 in a murine B cell line.** Growth curves of CR6 and CR6$^{F514I}$ in M12 cells over 4 days, inoculated at an MOI of 0.05. Results combined from three independent experiments, analyzed by repeated-measures two-way ANOVA, with Šídák's multiple comparison test. a indicates significant difference vs. CR6 ($P < 0.05$); b indicates significant difference vs. CR6 and CR6$^{F514I}$ ($P < 0.05$).
(TIF)

**S9 Fig. Potential interactions of VP1 residue 514 in MNoV S7 and CW3 capsid structures.** **(A)** Aligned sequences of MNoV S7 and CW3 VP1 proteins showing sequence conservation and residues with which amino acid 514 may interact. Yellow highlights indicate putative 514-interacting residues (within 4 angstroms, as predicted by UCSF Chimera); grey highlight indicates residue 514. Bars above each row of sequence indicate the domain, colored as in **Fig 8**, with the shell, P1, and P2 domains in cyan, white, and magenta, respectively, and with the linker region (residues 210–236) additionally shown in dark blue. Alignment performed using Clustal Omega. **(B)** A view of the structure of MNoV S7 VP1, showing residue F514 (in green) and nearby residues. Domains are colored as in panel A, with residues with which F514 may interact (as listed in **(A)**) colored in yellow, with side chains additionally shown for those amino acids. **(C)** A view of the structure of MNoV CW3 VP1, with coloring and visualizations done as for S7 in **(B)**. UCSF Chimera used for all visualizations.
(TIF)

**S1 Table. Shared mutations at specific nucleotides across the MNoV genome were observed when passaging CR6 *in vivo*.** Positions throughout the MNoV CR6 genome at which a mutation was observed in more than one mouse are shown. Nucleotides are labeled by position within the genome and the protein(s) within which the mutation is found. Samples are sorted by mouse genotype and round, followed by dpi. Total sequencing reads, percentage of reads identified as MNoV, and average coverage across the MNoV genome are shown for each sample. For each mutation, individual samples are labeled in grey (indicating insufficient coverage at that position to call a mutation at that nucleotide position), yellow (sufficient coverage, and the consensus sequence of that base agrees with the reference), or red (sufficient coverage, with the consensus base differing from the reference, and with the percentage of reads differing from the reference shown in each square).
(TIF)

## Acknowledgments

We acknowledge all members of the Baldridge laboratory for helpful discussions. We also thank H. Deng and L. Foster for assistance with animal care and breeding, and X. Zhang for technical assistance.

## Author Contributions

**Conceptualization:** Broc T. McCune, Stephanie M. Karst, Timothy J. Nice, Craig B. Wilen, Megan T. Baldridge.

**Funding acquisition:** Craig B. Wilen, Megan T. Baldridge.

**Investigation:** Forrest C. Walker, Ebrahim Hassan, Stefan T. Peterson, Lawrence A. Schriefer, Cassandra E. Thompson, Yuhao Li, Carla Blum-Johnston, Vincent R. Graziano, Larissa Lushniak, Alexa N. Roth, Megan T. Baldridge.

**Methodology:** Forrest C. Walker, Ebrahim Hassan, Stefan T. Peterson, Lawrence A. Schriefer, Jonathan J. Miner.

**Software:** Rachel Rodgers, Dylan Lawrence.

**Supervision:** Stephanie M. Karst, Craig B. Wilen, Megan T. Baldridge.

**Visualization:** Forrest C. Walker, Rachel Rodgers, Dylan Lawrence.

**Writing – original draft:** Forrest C. Walker, Gowri Kalugotla, Megan T. Baldridge.

**Writing – review & editing:** Ebrahim Hassan, Sanghyun Lee, Stephanie M. Karst, Timothy J. Nice, Craig B. Wilen, Megan T. Baldridge.

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
