## [Decision Letter · Decision Letter 0]

9 Nov 2020

Dear Dr Baldridge,

Thank you very much for submitting your manuscript "Norovirus evolution in immunodeficient mice reveals potentiated pathogenicity via a single nucleotide change in the viral capsid" for consideration at PLOS Pathogens. As with all papers reviewed by the journal, your manuscript was reviewed by members of the editorial board and by several independent reviewers. In light of the reviews (below this email), we would like to invite the resubmission of a significantly-revised version that takes into account the reviewers' comments.

The reviewers appreciated the attention to an interesting topic and the high quality of the work. A number of modifications aimed at increasing the clarity of the text and figures were suggested, and we encourage you to consider this feedback carefully. In addition, Reviewer 3 raised concerns that the conclusions drawn from Figure 7 are not well supported. Namely, the conclusion that I514 increases systemic virus spread due to increased monocyte recruitment early during infection was questioned. In light of this, we ask that you consider the reviewer's suggested strategies for enhanced analysis of the existing data and follow up with additional experiments if needed. Please provide a point-by-point response that addresses all reviewers' comments and ensure that concerns are also addressed within the manuscript text. 

We cannot make any decision about publication until we have seen the revised manuscript and your response to the reviewers' comments. Your revised manuscript may also be sent to reviewers for further evaluation.

Sincerely,

Anice C. Lowen

Associate Editor

PLOS Pathogens

Christopher Basler

Section Editor

PLOS Pathogens

Kasturi Haldar

Editor-in-Chief

PLOS Pathogens

orcid.org/0000-0001-5065-158X

Michael Malim

Editor-in-Chief

PLOS Pathogens

orcid.org/0000-0002-7699-2064

The reviewers appreciated the attention to an interesting topic and the high quality of the work. A number of modifications aimed at increasing the clarity of the text and figures were suggested, and we encourage you to consider this feedback carefully. In addition, Reviewer 3 raised concerns that the conclusions drawn from Figure 7 are not well supported. Namely, the conclusion that I514 increases systemic virus spread due to increased monocyte recruitment early during infection was questioned. In light of this, we ask that you consider the reviewer's suggested strategies for enhanced analysis of the existing data and follow up with additional experiments if needed.

Reviewer's Responses to Questions

**Part I - Summary**

Reviewer #1: This study describes the identification of a single amino acid mutation in the capsid protein that leads to the conversion of an avirulent strain of MNoV to a virulent phenotype in immunocompromised mice. The manuscript describes a rather complex set of experiments due to the combination of different viral genotypes, inoculation routes, time points, and mouse stains. Division of the results section into discrete sections for different set of experiments, each with a sentence at the end of each section that summarized the findings was extremely helpful. Only minor revisions are suggested by this reviewer:

Data were illustrated in numerous figures for each section and because the results are described in the text of each section it might be helpful to move a subset of the figures into a supplementary data section (i.e. Fig 3B and 3E, Fig 4 Sanger traces) to make the paper easier to follow.

The mouse strain phenotypes used should be briefly described when first mentioned in the manuscript (line 82 and line 353).

Line 180: Why was a 1:80 dilution selected for passage of brain homogenate?

Lines 200-209, Suppl. 3A, etc. Data from the deep sequencing is not clearly described. For example, is it known why there is so much variation between samples in sequencing depth across each genome? Were other high frequency variants detected in > 1 sample?

Line 293: One of 3 mice were analyzed for the emergence of F514I; given the relative ease of Sanger sequencing it might be informative to check additional mice for F514I emergence.

Fig 8 A & B. Without the labeling of the protein domains and the mutation site it is hard to understand what these pictures are illustrating.

A short paragraph in the discussion section on how this MNoV model helps us understand human norovirus disease would increase the general relevance of the finding. For example, do human noroviruses evolve differently in human hosts that are immunocompromised (i.e. due to chronic infections or malnutrition)?

Reviewer #2: In the manuscript by Walker et al., the authors investigate the effect of serial growth of a non-lethal strain of murine norovirus in both WT and Stat1-/- mice. They identify the emergence of a single mutation in the capsid that is able to confer lethality upon oral inoculation to the normally non-lethal CR6 strain (F514I), and show that introducing the reverse mutation into the lethal strain CW3 shows largely the opposite effect. In a series of detailed experiments, the authors are able to show that the effect of the mutation on lethality is associated with an increased ability to spread, while this mutation incurs a fitness cost in WT mice, where reduced viral loads are observed in the intestine. Moreover, the authors show that this mutation results in higher viral loads and spread when Stat1 is non-functional in hematopoietic cells, implicating these as responsible for the spread, and confirm that virus titers are higher in the blood of infected Stat1-/- mice (both serum and cell fraction) infected with the F514I variant.

Overall, the work is very exhaustive and performed to a high standard. It conclusively shows the effect of this mutation on pathogenesis in vivo and provides insight into the potential mechanism underlying the effect. Moreover, it shows that a single capsid mutation can have important consequences for pathogenesis, as has been seen in other models (e.g. coxsackievirus B3). I find the fact that this mutation was observed in the receptor knock out mice lacking Stat1 (in a previous Plos Path publication from the authors) very interesting. The connection between an alternative receptor and pathogenesis of this mutation remains to be explored (and its relationship to hematopoietic cells).

Reviewer #3: Manuscript PPATHOGENS-D-20-02168, entitled Norovirus evolution in immunodeficient mice reveals potentiated pathogenicity via a single nucleotide change in the viral capsid describes studies aimed broadly at understanding why murine norovirus (MNV) strains display essentially two different, dramatically distinct, in vivo phenotypes. The first phenotype is typified by the CR6 strain that exhibits persistent nonlethal infection of mice that is largely restricted to the GI tract after oral inoculation. The second phenotype is typified by the CW3 strain that is acute in WT mice but establishes a wide-ranging systemic infection in IFN-deficient mice, which is rapidly lethal. The CW3-like viruses were the first isolated, which was through intracranial passage of MVN in mice lacking IFN signaling. Thus, the authors ask whether/how exposure of nonlethal viruses like CR6 might gain mutations that increase virulence in immunocompromised mice. The authors found that CR6 inoculation intracranially into Stat1ko mice resulted in the emergence of virus progeny that exhibit increased virulence after oral inoculation. This was due to a polymorphism in the VP1 protein resulting in a F514I amino acid change. This type of amino acid change correlates with the persistent vs. acute phenotype. A recombinant CR6 bearing this single amino acid mutation was sufficient to confer the more virulent, systemic infection phenotype. The CR6-F514I mutation resulted in greater replication largely in inflammatory monocytes and a phenotype for replication in the MLN and PP that are somewhat intermediate between parental CR6 and the acute CW3 strain.

**Part II – Major Issues: Key Experiments Required for Acceptance**

Reviewer #1: None

Reviewer #2: None

Reviewer #3: Overall, this manuscript is well-written and founded on an interesting scientific premise. The experiments are properly designed and the results are tantalizing. Data showing that the F514I mutation confers a more virulent phenotype similar to acute strains is convincing. The less developed extension of this observation to the L514 polymorphism is less convincing, mainly because this used an isolate and not a recombinant virus with a single polymorphism. The major weakness in the paper is in the authors’ conclusion that the I514 change increases systemic virus spread due to increased monocyte recruitment early during infection, similar to CW3. The authors’ conclusion may ultimately be correct; however, the data presented in Fig 7 are not strong enough to support this conclusion. The point-by-point analysis below may aid in strengthening these data:

1) Fig 7A,B: The authors quantitate the number of total and MNV-infected monocytes in the MLN and PP at 2 dpi. In Fig 7A, they conclude that CW3 infection exhibits increased monocytes in the MLN and PP, which is consistent with a previous report and supported by statistical analysis. However, they further conclude that CW3 and CR6-F514I exhibited similar levels of monocyte recruitment in the PP. This is based on the a non-significant difference between CW3 and CR6-F514I. This is based on using CW3 as the comparator; however, it seems more appropriate to use CR6 as the comparator, as it is the polymorphism in CR6 that is being tested. However, the most important comparison is whether CR6-F514I confers increased % monocytes in the PP. That statistical comparison is not provided but is crucial to support their conclusions. re are two problems with this data: (1) the authors show that monocyte levels are high for CW3 than for CR6, but did not do the same analysis for CR6-F514I.

Additionally, the in Fig 7B, the authors conclude that there is a trend for an increased number of MNV infected monocytes in the MLN of CR6-F514I than in those infected with CR6. However, the is not a clear trend and is strongly driven by a high degree of animal-to-animal variability. Given the authors’ conclusion that I514 increases monocyte recruitment, the experimental evidence must be stronger than a trend.

2) Fig 7C: Another facet of the authors’ conclusion is that the monocyte recruitment is an early event during infection that is driven by a shift in tropism from Tuft cells to monocytes in the Peyer’s Patch. This is tested by measuring the viral genome copies in ileum, colon and Peyer’s Patches on 1 dpi. As for 5 dpi, the results show a clear decrease in genome copies in ileum and colon. However, in the Peyer’s Patch the variability in the data renders the difference non-significant. While this trend is more convincing than that claimed in Fig 7B, the veracity of the conclusion is weakened by this data.

Overall, in Fig 7A-C, there is considerable variability that confounds clear interpretation of the results. Given the variability between individual animals, it would be reasonable to filter the data with an Outlier statistical analysis. Either Grubbs or ROUT would be acceptable for this. This may require additional animal numbers to maintain power for the statistical analyses.

Other major questions:

1) While the authors’ work has convinced me that the F514I variant confers a replication advantage in monocytes (and other lymphoid tissue), the bigger questions that are not well address is why and how? As part of the major capsid protein, it is natural to speculate that it involves entry. But then why would entry be more efficient in monocytes than in Tuft cells. Is there any way the nucleotide polymorphism could have its affect at the RNA level rather than the protein level?

2) Is passage in the brain preferential for the F514I variant? Does it also arise quickly after IP or IV inoculation? Does the F514I variant arise even at low levels by in vitro propagation of virus in BV2 cells? Are the IC passage experiments showing a true de novo mutation arising, or is it establishment of a minor variant as the major species due to a considerable replication advantage? Similar, will the F514I variant emerge from any immunodeficient model or is it an exclusive feature of IFN-signaling deficient animals.

3) Since F514I seems to arise frequently. Does having this variant as part of the quasispecies benefit parental CR6 in some way?

4) The data in Fig 5 are interesting and clearly show a replication detriment in CW3 with an I514F mutation. However, is it surprising that CW3 doesn’t replicate more like CR6?

5) In the Tuft cells studies (Fig 6), the mice infected with CR6-F514I exhibited ~50% survival in both the Pou2f3+/- and Pou2f3-/- mice on the STAT1-/- background; however, in the previous Fig there was only ~15-20% survival in STAT1-/- mice. This seems like a large difference given a largely similar genetic background. What is driving this difference?

**Part III – Minor Issues: Editorial and Data Presentation Modifications**

Reviewer #1: Raw deep sequencing data should be deposited in publicly available database.

Fig 8 A & B. Without the labeling of the protein domains and the mutation site it is hard to understand what these pictures are illustrating.

Reviewer #2: Comments:

1. It is not clear to me how easy it would have been to pick up this mutation from sequence alignments, as it appears in figure 2D. It would be helpful to know if multiple mutations are conserved in acute versus persistent infections in P1 or P2. Including such alignment in the supplement would be helpful.

2. It would be helpful to have a table summarizing mutations that appear at high frequency in the NGS and their frequency assuming sufficient coverage is available. I can’t tell right now how frequent the F514I mutation was in the NGS data (the Sanger suggests this mutation is at 100%), nor the existence of additional mutations that could be of interest. Also, what is the y axis in figure 3A.

3. Figure S2B/C should indicate how well the curve fits the data, not just the p value (R2).

4. Line 243 says no differences associated with the mutation, but there is a difference in stat1-/- mice in the MLN (figure 3F).

5. It could be interesting to see if the mutant virus is better at infecting hematopoietic cells in vitro than the WT virus from stat1 knock out or WT mice . However, the experiment in figure 7D provides good evidence to suggest this.

Minor comments:

1. Missing words in the sentences on lines 87 and 192

2. Define set1 in the text

3. Line 310, white square should be blue square

4. NGS accession codes are missing.

Reviewer #3: Minor comments:

Lines 87-89: Could also contrast the limited genotype diversity among MNV with the high diversity among HuNoV.

Lines 127-128: The last sentence should be revised to be more precise. What elements of pathogenesis have you revealed?

Fig 1C: Consider a rainbow heatmap. The single color, particularly red, is very difficult to visualize the small gradations.

Lines 239-240: This sentence reads awkwardly to me. Perhaps: “When administered to Stat-/- mice PO, CR6-F514I induced significant weight loss, as well as greater splenomegaly and lethality than CR6”

**Note, in Fig 3C there is no statistical analysis of spleen weight between CR6 and CR6-F514I. This needs to be added because the comparison is written into the text in lines 239-240.

Lines 274-275: read awkward. Perhaps: “only a limited cohort. Surviving mice regained some weight but remained underweight compared to other cohorts.”

Fig 5: Perhaps change the diamond symbol to another shape. It is difficult to differentiate the circle and diamond, and this will get more difficult when the figure is reduced in size for publication.

PLOS authors have the option to publish the peer review history of their article (what does this mean?). If published, this will include your full peer review and any attached files.

Reviewer #1: No

Reviewer #2: No

Reviewer #3: **Yes: **Joseph Hyser
---

## [Decision Letter · Decision Letter 1]

17 Feb 2021

Dear Dr Baldridge,

We are pleased to inform you that your manuscript 'Norovirus evolution in immunodeficient mice reveals potentiated pathogenicity via a single nucleotide change in the viral capsid' has been provisionally accepted for publication in PLOS Pathogens.

Note that Reviewer 2 has made a couple of suggestions about very minor formatting modifications that you may wish to consider.

Best regards,

Anice C. Lowen

Associate Editor

PLOS Pathogens

Christopher Basler

Section Editor

PLOS Pathogens

Kasturi Haldar

Editor-in-Chief

PLOS Pathogens

orcid.org/0000-0001-5065-158X

Michael Malim

Editor-in-Chief

PLOS Pathogens

orcid.org/0000-0002-7699-2064

Reviewer Comments (if any, and for reference):

Reviewer's Responses to Questions

**Part I - Summary**

Reviewer #2: I feel that the modified version has addressed the major changes that were requested.

Reviewer #3: My apologies for brevity, this reviewer is without power and writing the review on a phone.

In depth study.

Very strong responses to reviewers’ comments

Weaknesses or contradictions in the data from different experiments are addressed in the text

**Part II – Major Issues: Key Experiments Required for Acceptance**

Reviewer #2: None

Reviewer #3: No further major issues

**Part III – Minor Issues: Editorial and Data Presentation Modifications**

Reviewer #2: 1. Since the paper deals a lot with CR6, it would be useful to include this in the table in Figure S3D.

2. Line 84, "innate IFN signaling" should probably be innate-immunity

3. In many places there are mentions of results not being significant (e.g. Fig. 6B,C,D). It would be nice to actually show this on the graph, and to put the p-value so that one can judge how far away the result is from the somewhat arbitrary p<0.05 cutoff. I realize this is a mess when there are many possible comparisons, but for the 2 sample comparisons this could be helpful. I would potentially include this for relevant comparisons in fig. 7 A/B.

Reviewer #3: No further minor issues.

PLOS authors have the option to publish the peer review history of their article (what does this mean?). If published, this will include your full peer review and any attached files.

Reviewer #2: No

Reviewer #3: No

---

## [Editor Report · Acceptance letter]

8 Mar 2021

Dear Dr Baldridge,

We are delighted to inform you that your manuscript, "Norovirus evolution in immunodeficient mice reveals potentiated pathogenicity via a single nucleotide change in the viral capsid," has been formally accepted for publication in PLOS Pathogens.

Best regards,

Kasturi Haldar

Editor-in-Chief

PLOS Pathogens

orcid.org/0000-0001-5065-158X

Michael Malim

Editor-in-Chief

PLOS Pathogens

orcid.org/0000-0002-7699-2064